# SOLUTION: BYZANTINE CLUSTER-SENDING IN EXPECTED CONSTANT COST AND CONSTANT TIME

## Abstract

Traditional resilient systems operate on fully-replicated fault-tolerant clusters, which limits their scalability and performance. One way to make the step towards resilient high-performance systems that can deal with huge workloads, is by enabling independent fault-tolerant clusters to efficiently communicate and cooperate with each other, as this also enables the usage of high-performance techniques such as sharding. Recently, such inter-cluster communication was formalized as the *Byzantine cluster-sending problem*. Unfortunately, existing worst-case optimal protocols for cluster-sending all have *linear complexity* in the size of the clusters involved.

In this paper, we propose *probabilistic cluster-sending techniques* as a solution for the cluster-sending problem with only an *expected constant message complexity*, this independent of the size of the clusters involved and this even in the presence of highly unreliable communication. Depending on the robustness of the clusters involved, our techniques require only *two-to-four* message round-trips (without communication failures). Furthermore, our protocols can support worst-case linear communication between clusters. Finally, we have put our techniques to the test in an in-depth experimental evaluation that further underlines the exceptional low expected costs of our techniques in comparison with other protocols. As such, our work provides a strong foundation for the further development of resilient high-performance systems.

## 1 Introduction

The promises of *resilient data processing*, as provided by private and public blockchains [14, 20, 26], has renewed interest in traditional consensus-based Byzantine fault-tolerant resilient systems [5, 6, 23]. Unfortunately, blockchains and other consensus-based systems typically rely on fully-replicated designs, which limits their scalability and performance. Consequently, these systems cannot deal with the ever-growing requirements in data processing [28, 29].

One wat to improve on these limitations is by building complex system designs that consist of *independently-operating* resilient clusters that can cooperate to provide certain services. To illustrate this, one can consider a sharded resilient design. In a traditional resilient systems, resilience is provided by a fully-replicated consensus-based Byzantine fault-tolerant cluster in which all replicas hold all data and process all requests. This traditional design has only limited performance,

even with the best consensus protocols, and lacks scalability. To improve on the design of traditional systems, one can employ the *sharded* design of Figure 1. In this sharded design, each cluster only holds part of the data. Consequently, each cluster only needs to process requests that affect data they hold. In this way, this sharded design improves performance by enabling *parallel processing* of requests by different clusters, while also improving storage scalability. To support requests that affect data in several clusters in such a sharded design, the clusters need to be able to *coordinate their operations*, however [1, 7, 15, 18].

Central to such complex system designs is the ability to reliably and efficiently communicate between independently-operating resilient clusters. Recently, this problem of communication *between* Byzantine fault-tolerant clusters has been formalized as the *cluster-sending problem* [17]. We believe that efficient solutions to this problem have a central role towards bridging *resilient* and *high-performance* data processing.

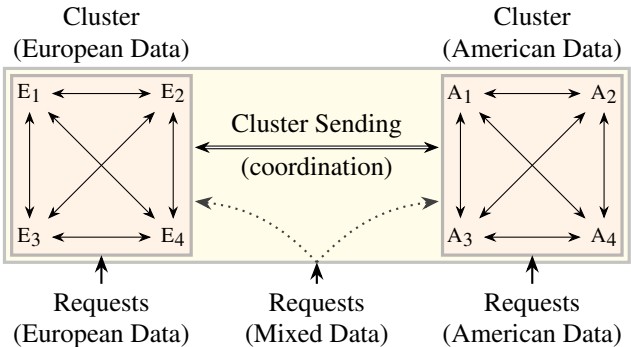

Figure 1: A *sharded* design in which each resilient cluster of four replicas holds only a part of the data. Local decisions within a cluster are made via *consensus* (⟷), whereas multi-shard coordination to process multi-shard transactions requires *cluster-sending* (⟺).

Although the cluster-sending problem has received some attention (e.g., as part of the design of AHL [7], BYSHARD [18], GEOBFT [15], and CHAINSPACE [1]), and cluster-sending protocols that solve the cluster-sending problem with worst-case optimal complexity are known [17], we believe there is still much room for improvement.

In this paper, we introduce a new solution to the cluster-

Figure 2: A comparison of *cluster-sending protocols* that send a value from cluster $C_1$ with $\mathbf{n}_{C_1}$ replicas, of which $\mathbf{f}_{C_1}$ are faulty, to cluster $C_2$ with $\mathbf{n}_{C_2}$ replicas, of which $\mathbf{f}_{C_2}$ are faulty. For each protocol $P$, *Protocol* specifies its name; *Robustness* specifies the conditions $P$ puts on the clusters; *Message Steps* specifies the number of messages exchanges $P$ performs; *Optimal* specifies whether $P$ is worst-case optimal; and *Unreliable* specifies whether $P$ can deal with unreliable communication.

| | Protocol | Robustness[a] | Message Steps (expected-case) | (worst-case) | Optimal | Unreliable |
|---|---|---|---|---|---|---|
| | PBS-CS [17] | $\min(\mathbf{n}_{C_1}, \mathbf{n}_{C_2}) > \mathbf{f}_{C_1} + \mathbf{f}_{C_2}$ | $\mathbf{f}_{C_1} + \mathbf{f}_{C_2} + 1$ | | ✔ | ✘ |
| | PBS-CS [17] | $\mathbf{n}_{C_1} > 3\mathbf{f}_{C_1}, \mathbf{n}_{C_2} > 3\mathbf{f}_{C_2}$ | $\max(\mathbf{n}_{C_1}, \mathbf{n}_{C_2})$ | | ✔ | ✘ |
| | GEOBFT [15] | $\mathbf{n}_{C_1} = \mathbf{n}_{C_2} > 3\max(\mathbf{f}_{C_1}, \mathbf{f}_{C_2})$ | $\mathbf{f}_{C_2} + 1$[b] | $\Omega(\mathbf{f}_{C_1}\mathbf{n}_{C_2})$ | ✘ | ✔ |
| | CHAINSPACE [1] | $\mathbf{n}_{C_1} > 3\mathbf{f}_{C_1}, \mathbf{n}_{C_2} > 3\mathbf{f}_{C_2}$ | $\mathbf{n}_{C_1}\mathbf{n}_{C_2}$ | | ✘ | ✘ |
| This Paper | CSPP | $\mathbf{n}_{C_1} > 2\mathbf{f}_{C_1}, \mathbf{n}_{C_2} > 2\mathbf{f}_{C_2}$ | 4 | $(\mathbf{f}_{C_1} + 1)(\mathbf{f}_{C_2} + 1)$ | ✘ | ✔ |
| | CSPP | $\mathbf{n}_{C_1} > 3\mathbf{f}_{C_1}, \mathbf{n}_{C_2} > 3\mathbf{f}_{C_2}$ | $2\frac{1}{4}$ | $(\mathbf{f}_{C_1} + 1)(\mathbf{f}_{C_2} + 1)$ | ✘ | ✔ |
| | CSPL | $\min(\mathbf{n}_{C_1}, \mathbf{n}_{C_2}) > \mathbf{f}_{C_1} + \mathbf{f}_{C_2}$ | 4 | $\mathbf{f}_{C_1} + \mathbf{f}_{C_2} + 1$ | ✔ | ✔ |
| | CSPL | $\min(\mathbf{n}_{C_1}, \mathbf{n}_{C_2}) > 2(\mathbf{f}_{C_1} + \mathbf{f}_{C_2})$ | $2\frac{1}{4}$ | $\mathbf{f}_{C_1} + \mathbf{f}_{C_2} + 1$ | ✔ | ✔ |
| | CSPL | $\mathbf{n}_{C_1} > 3\mathbf{f}_{C_1}, \mathbf{n}_{C_2} > 3\mathbf{f}_{C_2}$ | 3 | $\max(\mathbf{n}_{C_1}, \mathbf{n}_{C_2})$ | ✔ | ✔ |

[a]Protocols that have different message step complexities depending on the robustness assumptions have been included for each of the robustness assumptions.
[b]Complexity when the coordinating primary in $C_1$ is non-faulty and communication is reliable.

sending problem: we introduce cluster-sending protocols that use *probabilistic cluster-sending* techniques and are able to provide low *expected-case* message complexity (at the cost of higher communication latencies, a good trade-off in systems where inter-cluster network bandwidth is limited). In specific, our main contributions are as follows:

1. First, in Section 3, we introduce the cluster-sending step CS-STEP that attempts to send a value from a replica in the sending cluster to a replica in the receiving cluster in a verifiable manner and with a constant amount of inter-cluster communication.

2. Then, in Section 4, we introduce the *Synchronous Probabilistic Cluster-Sending protocol* CSP that uses CS-STEP with randomly selected sending and receiving replicas to provide cluster-sending in *expected constant* steps. We also propose *pruned*-CSP (CSPP), a fine-tuned version of CSP that guarantees termination.

3. In Section 5, we propose the *Synchronous Probabilistic Linear Cluster-Sending protocol* CSPL, that uses CS-STEP with a specialized randomized scheme to select replicas, this to provide cluster-sending in *expected constant* steps and *worst-case linear* steps, which is optimal.

4. Next, in Section 6, we discuss how CSP, CSPP, and CSPL can be generalized to operate in environments with *asynchronous and unreliable communication*.

5. Finally, in Section 7, we evaluate the behavior of the proposed probabilistic cluster-sending protocols via an in-depth evaluation. In this evaluation, we show that probabilistic cluster-sending protocols has exceptionally low communication costs in comparison with existing

cluster-sending protocols, this even in the presence of communication failures.

A summary of our findings in comparison with existing techniques can be found in Figure 2. In Section 2, we introduce the necessary terminology and notation, in Section 8, we compare with related work, and in Section 9, we conclude on our findings.

## 2 The Cluster-Sending Problem

Before we present our probabilistic cluster-sending techniques, we first introduce all necessary terminology and notation. The formal model we use is based on the formalization of the cluster-sending problem provided by Hellings et al. [17]. If $S$ is a set of replicas, then $\mathsf{f}(S) \subseteq S$ denotes the *faulty replicas* in $S$, whereas $\mathsf{nf}(S) = S \setminus \mathsf{f}(S)$ denotes the *non-faulty replicas* in $S$. We write $\mathbf{n}_S = |S|$, $\mathbf{f}_S = |\mathsf{f}(S)|$, and $\mathbf{nf}_S = |\mathsf{nf}(S)| = \mathbf{n}_S - \mathbf{f}_S$ to denote the number of replicas, faulty replicas, and non-faulty replicas in $S$, respectively. A *cluster* $C$ is a finite set of replicas. We consider clusters with *Byzantine replicas* that behave in arbitrary manners. In specific, if $C$ is a cluster, then any malicious adversary can control the replicas in $\mathsf{f}(C)$ at any time, but adversaries cannot bring non-faulty replicas under their control.

**Definition 2.1.** Let $C_1, C_2$ be disjoint clusters. The *cluster-sending problem* is the problem of sending a value $v$ from $C_1$ to $C_2$ such that **(1)** all non-faulty replicas in $\mathsf{nf}(C_2)$ RECEIVE the value $v$; **(2)** all non-faulty replicas in $\mathsf{nf}(C_1)$ CONFIRM that the value $v$ was received by all non-faulty replicas in $\mathsf{nf}(C_2)$; and **(3)** non-faulty replicas in $\mathsf{nf}(C_2)$ only receive a value $v$ if all non-faulty replicas in $\mathsf{nf}(C_1)$ AGREE upon sending $v$.

We assume that there is no limitation on local communication within a cluster, while global communication between

clusters is costly. This model is supported by practice, where communication between wide-area deployments of clusters is up-to-two orders of magnitudes more expensive than communication within a cluster [7, 15].

We assume that each cluster can make *local decisions* among all non-faulty replicas, e.g., via a *consensus protocol* such as PBFT or PAXOS [6, 23]. Furthermore, we assume that the replicas in each cluster can certify such local decisions via a *signature scheme*. E.g., a cluster $\mathcal{C}$ can certify a consensus decision on some message $m$ by collecting a set of signatures for $m$ of $\mathbf{f}_C + 1$ replicas in $\mathcal{C}$, guaranteeing one such signature is from a non-faulty replica (which would only signs values on which consensus is reached). We write $\langle m \rangle_C$ to denote a message $m$ certified by $\mathcal{C}$. To minimize the size of certified messages, one can utilize a threshold signature scheme [30]. To enable decision making and message certification, we assume, for every cluster $\mathcal{C}$, $\mathbf{n}_C > 2\mathbf{f}_C$, a minimal requirement [9, 24]. Lastly, we assume that there is a common source of randomness for all non-faulty replicas of each cluster, e.g., via a distributed fault-tolerant random coin [3, 4].

## 3  The Cluster-Sending Step

If communication is reliable and one knows non-faulty replicas $R_1 \in \mathsf{nf}(\mathcal{C}_1)$ and $R_2 \in \mathsf{nf}(\mathcal{C}_2)$, then cluster-sending a value $v$ from $\mathcal{C}_1$ to $\mathcal{C}_2$ can be done via a straightforward *cluster-sending step*: one can simply instruct $R_1$ to send $v$ to $R_2$. When $R_2$ receives $v$, it can disperse $v$ locally in $\mathcal{C}_2$. Unfortunately, we do not know which replicas are faulty and which are non-faulty. Furthermore, it is practically impossible to reliably determine which replicas are non-faulty, as non-faulty replicas can appear faulty due to unreliable communication, while faulty replicas can appear well-behaved to most replicas, while interfering with the operations of only some non-faulty replicas.

To deal with faulty replicas when utilizing the above *cluster-sending step*, one needs a sufficient safeguards to detect *failure* of $R_1$, of $R_2$, or of the communication between them. To do so, we add receive and confirmation phases to the sketched cluster-sending step. During the *receive phase*, the receiving replica $R_2$ must construct a proof $P$ that it received and dispersed $v$ locally in $\mathcal{C}_2$ and then send this proof back to $R_1$. Finally, during the *confirmation phase*, $R_1$ can utilize $P$ to prove to all other replicas in $\mathcal{C}_1$ that the cluster-sending step was successful. The pseudo-code of this *cluster-sending step protocol* CS-STEP can be found in Figure 3. We have the following:

**Proposition 3.1.** *Let $\mathcal{C}_1, \mathcal{C}_2$ be disjoint clusters with $R_1 \in \mathcal{C}_1$ and $R_2 \in \mathcal{C}_2$. If $\mathcal{C}_1$ satisfies the pre-conditions of* CS-STEP($R_1$, $R_2$, $v$)*, then execution of* CS-STEP($R_1$, $R_2$, $v$) *satisfies the post-conditions and will exchange at most two messages between $\mathcal{C}_1$ and $\mathcal{C}_2$.*

*Proof.* We prove the three post-conditions separately. **(i)**

---

**Protocol** CS-STEP($R_1$, $R_2$, $v$), with $R_1 \in \mathcal{C}_1$ and $R_2 \in \mathcal{C}_2$:

**Pre:** Each replica in $\mathsf{nf}(\mathcal{C}_1)$ decided AGREE on sending $v$ to $\mathcal{C}_2$ (and can construct $\langle \mathtt{send} : v, \mathcal{C}_2 \rangle_{\mathcal{C}_1}$).

**Post:** **(i)** If communication is reliable, $R_1 \in \mathsf{nf}(\mathcal{C}_1)$, and $R_2 \in \mathsf{nf}(\mathcal{C}_2)$, then $R_1$ decides CONFIRM on $v$. **(ii)** If a replica in $\mathsf{nf}(\mathcal{C}_2)$ decides RECEIVE on $v$, then all replicas in $\mathsf{nf}(\mathcal{C}_1)$ decided AGREE on sending $v$ to $\mathcal{C}_2$. **(iii)** If a replica in $\mathsf{nf}(\mathcal{C}_1)$ decides CONFIRM on $v$, then all replicas in $\mathsf{nf}(\mathcal{C}_2)$ decided RECEIVE on $v$ and all replicas in $\mathsf{nf}(\mathcal{C}_1)$ eventually decide CONFIRM on $v$ (whenever communication becomes reliable).

**The *cluster-sending step* for $R_1$ and $R_2$:**

1: Instruct $R_1$ to send $\langle \mathtt{send} : v, \mathcal{C}_2 \rangle_{\mathcal{C}_1}$ to $R_2$.

**The *receive role* for $\mathcal{C}_2$:**

2: **event** $R_2 \in \mathsf{nf}(\mathcal{C}_2)$ receives message $m := \langle \mathtt{send} : v, \mathcal{C}_2 \rangle_{\mathcal{C}_1}$ from $R_1 \in \mathcal{C}_1$ **do**
3:   **if** $R_2$ does not have consensus on $m$ **then**
4:     Use *local consensus* on $m$ and construct $\langle \mathtt{proof} : m \rangle_{\mathcal{C}_2}$.
5:     {*Each replica in $\mathsf{nf}(\mathcal{C}_2)$ decides RECEIVE on $v$.*}
6:   Send $\langle \mathtt{proof} : m \rangle_{\mathcal{C}_2}$ to $R_1$.

**The *confirmation role* for $\mathcal{C}_1$:**

7: **event** $R_1 \in \mathsf{nf}(\mathcal{C}_1)$ receives message $m_p := \langle \mathtt{proof} : m \rangle_{\mathcal{C}_2}$ with $m := \langle \mathtt{send} : v, \mathcal{C}_2 \rangle_{\mathcal{C}_1}$ from $R_2 \in \mathcal{C}_2$ **do**
8:   **if** $R_1$ does not have consensus on $m_p$ **then**
9:     Use *local consensus* on $m_p$.
10:     {*Each replica in $\mathsf{nf}(\mathcal{C}_1)$ decides CONFIRM on $v$.*}

---

Figure 3: The Cluster-sending step protocol CS-STEP($R_1$, $R_2$, $v$). In this protocol, $R_1$ tries to send $v$ to $R_2$, which will succeed if both $R_1$ and $R_2$ are non-faulty.

We assume that communication is reliable, $R_1 \in \mathsf{nf}(\mathcal{C}_1)$, and $R_2 \in \mathsf{nf}(\mathcal{C}_2)$. Hence, $R_1$ sends message $m := \langle \mathtt{send} : v, \mathcal{C}_2 \rangle_{\mathcal{C}_1}$ to $R_2$ (Line 1 of Figure 3). In the receive phase (Lines 2–6 of Figure 3), replica $R_2$ *receives* message $m$ from $R_1$. Replica $R_2$ uses local consensus on $m$ to replicate $m$ among all replicas $\mathcal{C}_2$ and, along the way, to constructs a *proof of receipt* $m_p := \langle \mathtt{proof} : m \rangle_{\mathcal{C}_2}$. As all replicas in $\mathsf{nf}(\mathcal{C}_2)$ participate in this local consensus, all replicas in $\mathsf{nf}(\mathcal{C}_2)$ will decide RECEIVE on $v$ from $\mathcal{C}_1$. Finally, the proof $m_p$ is returned to $R_1$. In the confirmation phase (Lines 7–10 of Figure 3), replica $R_1$ receives the proof of receipt $m_p$. Next, $R_1$ uses local consensus on $m_p$ to replicate $m_p$ among all replicas in $\mathsf{nf}(\mathcal{C}_1)$, after which all replicas in $\mathsf{nf}(\mathcal{C}_1)$ decide CONFIRM on sending $v$ to $\mathcal{C}_2$

**(ii)** A replica in $\mathsf{nf}(\mathcal{C}_2)$ only decides RECEIVE on $v$ after consensus is reached on a message $m := \langle \mathtt{send} : v, \mathcal{C}_2 \rangle_{\mathcal{C}_1}$ (Line 5 of Figure 3). This message $m$ not only contains the value $v$, but also the identity of the recipient cluster $\mathcal{C}_2$. Due to the usage of certificates and the pre-condition, the message $m$ cannot be created without the replicas in $\mathsf{nf}(\mathcal{C}_1)$ deciding AGREE on sending $v$ to $\mathcal{C}_2$.

**(iii)** A replica in $\mathsf{nf}(\mathcal{C}_1)$ only decides CONFIRM on $v$ after

consensus is reached on a *proof of receipt* message $m_p :=$ $\langle \texttt{proof} : m \rangle_{C_2}$ (Line 10 of Figure 3). This consensus step will complete for all replicas in $C_1$ whenever communication becomes reliable. Hence, all replicas in $\mathsf{nf}(C_1)$ will eventually decide CONFIRM on $v$. Due to the usage of certificates, the message $m_p$ cannot be created without cooperation of the replicas in $\mathsf{nf}(C_2)$. The replicas in $\mathsf{nf}(C_2)$ only cooperate in constructing $m_p$ as part of the consensus step of Line 4 of Figure 3. Upon completion of this consensus step, all replicas in $\mathsf{nf}(C_2)$ will decide RECEIVE on $v$. □

In the following sections, we show how to use the cluster-sending step in the construction of cluster-sending protocols. In Section 4, we introduce synchronous protocols that provide *expected constant message complexity*. Then, in Section 5, we introduce synchronous protocols that additionally provide *worst-case linear message complexity*, which is optimal. Finally, in Section 6, we show how to extend the presented techniques to asynchronous communication.

## 4 Probabilistic Cluster-Sending with Random Replica Selection

In the previous section, we introduced CS-STEP, the cluster-sending step protocol that succeeds whenever the participating replicas are non-faulty and communication is reliable. Using CS-STEP, we build a three-step protocol that cluster-sends a value $v$ from $C_1$ to $C_2$:

1. First, the replicas in $\mathsf{nf}(C_1)$ reach agreement and decide AGREE on sending $v$ to $C_2$.

2. Then, the replicas in $\mathsf{nf}(C_1)$ perform a *probabilistic cluster-sending step* by electing replicas $\mathtt{R_1} \in C_1$ and $\mathtt{R_2} \in C_2$ fully at random, after which CS-STEP($\mathtt{R_1}$, $\mathtt{R_2}$, $v$) is executed.

3. Finally, each replicas in $\mathsf{nf}(C_1)$ waits for the completion of CS-STEP($\mathtt{R_1}$, $\mathtt{R_2}$, $v$) If the waiting replicas decided CONFIRM on $v$ during this wait, then cluster-sending is successful. Otherwise, we repeat the previous step.

To simplify presentation, we assume *synchronous* inter-cluster communication to enable replicas to wait for completion: messages sent by non-faulty replicas will be delivered within some known bounded delay. We refer to Section 6 on how to deal with asynchronous and unreliable communication. *Synchronous* systems can be modeled by *pulses* [10, 11]:

**Definition 4.1.** A system is *synchronous* if all inter-cluster communication happens in *pulses* such that every message sent in a pulse will be received in the same pulse.

The pseudo-code of the resultant *Synchronous Probabilistic Cluster-Sending protocol* CSP can be found in Figure 4. Next, we prove that CSP is correct and has expected-case constant message complexity:

**Protocol** CSP($C_1$, $C_2$, $v$):

1: Use *local consensus* on $v$ and construct $\langle \texttt{send} : v, C_2 \rangle_{C_1}$.
2: {*Each replica in* $\mathsf{nf}(C_1)$ *decides* AGREE *on* $v$.}
3: **repeat**
4:     Choose replicas $(\mathtt{R_1}, \mathtt{R_2}) \in C_1 \times C_2$, fully at random.
5:     CS-STEP($\mathtt{R_1}$, $\mathtt{R_2}$, $v$)
6:     Wait *three* global pulses.
7: **until** $C_1$ reaches consensus on $\langle \texttt{proof} : \langle \texttt{send} : v, C_2 \rangle_{C_1} \rangle_{C_2}$.

Figure 4: The Synchronous Probabilistic Cluster-Sending protocol CSP($C_1$, $C_2$, $v$) that cluster-sends a value $v$ from $C_1$ to $C_2$.

**Theorem 4.2.** *Let $C_1, C_2$ be disjoint clusters. If communication is synchronous, then* CSP($C_1$, $C_2$, $v$) *results in cluster-sending $v$ from $C_1$ to $C_2$. The execution performs two local consensus steps in $C_1$, one local consensus step in $C_2$, and is expected to make $(\mathbf{n}_{C_1} \mathbf{n}_{C_2})/(\mathbf{nf}_{C_1} \mathbf{nf}_{C_1})$ cluster-sending steps.*

*Proof.* Due to Lines 1–2 of Figure 4, CSP($C_1$, $C_2$, $v$) establishes the pre-conditions for any execution of CS-STEP($\mathtt{R_1}$, $\mathtt{R_2}$, $v$) with $\mathtt{R_1} \in C_1$ and $\mathtt{R_2} \in C_2$. Using the correctness of CS-STEP (Proposition 3.1), we conclude that CSP($C_1$, $C_2$, $v$) results in cluster-sending $v$ from $C_1$ to $C_2$ whenever the replicas $(\mathtt{R_1}, \mathtt{R_2}) \in C_1 \times C_2$ chosen at Line 4 of Figure 4 are non-faulty. As the replicas $(\mathtt{R_1}, \mathtt{R_2}) \in C_1 \times C_2$ are chosen fully at random, we have probability $p_i = \mathbf{nf}_{C_i}/\mathbf{n}_{C_i}, i \in \{1, 2\}$, of choosing $\mathtt{R_i} \in \mathsf{nf}(C_i)$. The probabilities $p_1$ and $p_2$ are independent of each other. Consequently, the probability of choosing $(\mathtt{R_1}, \mathtt{R_2}) \in \mathsf{nf}(C_1) \times \mathsf{nf}(C_2)$ is $p = p_1 p_2 = (\mathbf{nf}_{C_1} \mathbf{nf}_{C_2})/(\mathbf{n}_{C_1} \mathbf{n}_{C_2})$. As such, each iteration of the loop at Line 3 of Figure 4 can be modeled as an independent *Bernoulli trial* with probability of success $p$, and the expected number of iterations of the loop is $p^{-1} = (\mathbf{n}_{C_1} \mathbf{n}_{C_2})/(\mathbf{nf}_{C_1} \mathbf{nf}_{C_1})$.

Finally, we prove that each local consensus step needs to be performed only once. To do so, we consider the local consensus steps triggered by the loop at Line 3 of Figure 4. These are the local consensus steps at Lines 4 and 9 of Figure 3. The local consensus step at Line 4 can be initiated by a faulty replica $\mathtt{R_2}$. After this single local consensus step reaches consensus on message $m := \langle \texttt{send} : v, C_2 \rangle_{C_1}$, each replica in $\mathsf{nf}(C_2)$ reaches consensus on $m$, decides RECEIVE on $v$, and can construct $m_p := \langle \texttt{proof} : m \rangle_{C_2}$, this independent of the behavior of $\mathtt{R_2}$. Hence, a single local consensus step for $m$ in $C_2$ suffices, and no replica in $\mathsf{nf}(C_2)$ will participate in future consensus steps for $m$. An analogous argument proves that a single local consensus step for $m_p$ in $C_1$, performed at Line 9 of Figure 3, suffices. □

*Remark* 4.3. Although Theorem 4.2 indicates local consensus steps in clusters $C_1$ and $C_2$, these local consensus steps typically come for *free* as part of the protocol that uses cluster-sending as a building block. To see this, we consider a multi-shard transaction processed by clusters $C_1$ and $C_2$.

The decision of cluster $C_1$ to send a value $v$ to cluster $C_2$ is a consequence of the execution of some transaction $\tau$ in $C_1$. Before the replicas in $C_1$ execute $\tau$, they need to reach consensus on the order in which $\tau$ is executed in $C_1$. As part of this consensus step, the replicas in $C_1$ can also construct $\langle \mathtt{send} : v, C_2 \rangle_{C_1}$ without additional consensus steps. Hence, no consensus step is necessary in $C_1$ to send value $v$. Likewise, if value $v$ is received by replicas in $C_2$ as part of some multi-shard transaction execution protocol, then the replicas in $C_2$ need to perform the necessary transaction execution steps as a *consequence* of receiving $v$. To do so, the replicas in $C_2$ need to reach consensus on the order in which these transaction execution steps are performed. As part of this consensus step, the replicas in $C_2$ can also constructing a proof of receipt for $v$.

In typical fault-tolerant clusters, at least half of the replicas are non-faulty (e.g., in synchronous systems with Byzantine failures that use digital signatures, or in systems that only deal with crashes) or at least two-third of the replicas are non-faulty (e.g., asynchronous systems). In these systems, CSP is expected to only performs a few cluster-sending steps:

**Corollary 4.4.** *Let $C_1, C_2$ be disjoint clusters. If communication is synchronous, then the expected number of cluster-sending steps performed by $\mathrm{CSP}(C_1, C_2, v)$ is upper bounded by 4 if $\mathbf{n}_{C_1} > 2\mathbf{f}_{C_1}$ and $\mathbf{n}_{C_2} > 2\mathbf{f}_{C_2}$; and by $2\frac{1}{4}$ if $\mathbf{n}_{C_1} > 3\mathbf{f}_{C_1}$ and $\mathbf{n}_{C_2} > 3\mathbf{f}_{C_2}$.*

In CSP, the replicas $(\mathtt{R}_1, \mathtt{R}_2) \in C_1 \times C_2$ are chosen fully at random and *with replacement*, as CSP does not retain any information on *failed* probabilistic steps. In the worst case, this prevents *termination*, as the same pair of replicas can be picked repeatedly. Furthermore, CSP does not prevent the choice of faulty replicas whose failure could be detected. We can easily improve on this, as the *failure* of a probabilistic step provides some information on the chosen replicas. In specific, we have the following technical properties:

**Lemma 4.1.** *Let $C_1, C_2$ be disjoint clusters. We assume synchronous communication and assume that each replica in $\mathsf{nf}(C_1)$ decided* AGREE *on sending $v$ to $C_2$.*

1. *Let $(\mathtt{R}_1, \mathtt{R}_2) \in C_1 \times C_2$. If CS-STEP($\mathtt{R}_1$, $\mathtt{R}_2$, $v$) fails to cluster-send $v$, then either $\mathtt{R}_1 \in \mathsf{f}(C_1)$, $\mathtt{R}_2 \in C_2$, or both.*

2. *Let $\mathtt{R}_1 \in C_1$. If CS-STEP($\mathtt{R}_1$, $\mathtt{R}_2$, $v$) fails to cluster-send $v$ for $\mathbf{f}_{C_2} + 1$ distinct replicas $\mathtt{R}_2 \in C_2$, then $\mathtt{R}_1 \in \mathsf{f}(C_1)$.*

3. *Let $\mathtt{R}_2 \in C_2$. If CS-STEP($\mathtt{R}_1$, $\mathtt{R}_2$, $v$) fails to cluster-send $v$ for $\mathbf{f}_{C_1} + 1$ distinct replicas $\mathtt{R}_1 \in C_1$, then $\mathtt{R}_2 \in \mathsf{f}(C_2)$.*

*Proof.* The statement of this Lemma assumes that the preconditions for any execution of CS-STEP($\mathtt{R}_1$, $\mathtt{R}_2$, $v$) with $\mathtt{R}_1 \in C_1$ and $\mathtt{R}_2 \in C_2$ are established. Hence, by Proposition 3.1, CS-STEP($\mathtt{R}_1$, $\mathtt{R}_2$, $v$) will cluster-send $v$ if $\mathtt{R}_1 \in \mathsf{nf}(C_1)$ and $\mathtt{R}_2 \in \mathsf{nf}(C_2)$. If the cluster-sending step fails to cluster-send

$v$, then one of the replicas involved must be faulty, proving the first property. Next, let $\mathtt{R}_1 \in C_1$ and consider a set $S \subseteq C_2$ of $\mathbf{n}_S = \mathbf{f}_{C_2} + 1$ replicas such that, for all $\mathtt{R}_2 \in S$, CS-STEP($\mathtt{R}_1$, $\mathtt{R}_2$, $v$) fails to cluster-send $v$. Let $S' = S \setminus \mathsf{f}(C_2)$ be the non-faulty replicas in $S$. As $\mathbf{n}_S > \mathbf{f}_{C_2}$, we have $\mathbf{n}_{S'} \geq 1$ and there exists a $\mathtt{R}_2' \in S'$. As $\mathtt{R}_2' \notin \mathsf{f}(C_2)$ and CS-STEP($\mathtt{R}_1$, $\mathtt{R}_2'$, $v$) fails to cluster-send $v$, we must have $\mathtt{R}_1 \in \mathsf{f}(C_1)$ by the first property, proving the second property. An analogous argument proves the third property. □

We can apply the properties of Lemma 4.1 to actively *prune* which replica pairs CSP considers (Line 4 of Figure 4). Notice that pruning via Lemma 4.1(1) simply replaces choosing replica pairs *with replacement*, as done by CSP, by choosing replica pairs *without replacement*, this without further reducing the possible search space. Pruning via Lemma 4.1(2) does reduce the search space, however, as each replica in $C_1$ will only be paired with a subset of $\mathbf{f}_{C_2} + 1$ replicas in $C_2$. Likewise, pruning via Lemma 4.1(3) also reduces the search space. We obtain the *Pruned Synchronous Probabilistic Cluster-Sending protocol* (CSPP) by applying all three prune steps to CSP. By construction, Theorem 4.2, and Lemma 4.1, we conclude:

**Corollary 4.5.** *Let $C_1, C_2$ be disjoint clusters. If communication is synchronous, then $\mathrm{CSPP}(C_1, C_2, v)$ results in cluster-sending $v$ from $C_1$ to $C_2$. The execution performs two local consensus steps in $C_1$, one local consensus step in $C_2$, is expected to make less than $(\mathbf{n}_{C_1} \mathbf{n}_{C_2})/(\mathbf{nf}_{C_1} \mathbf{nf}_{C_1})$ cluster-sending steps, and makes worst-case $(\mathbf{f}_{C_1} + 1)(\mathbf{f}_{C_2} + 1)$ cluster-sending steps.*

## 5 Worst-Case Linear-Time Probabilistic Cluster-Sending

In the previous section, we introduced CSP and CSPP, two probabilistic cluster-sending protocols that can cluster-send a value $v$ from $C_1$ to $C_2$ with expected constant cost. Unfortunately, CSP does not guarantee termination, while CSPP has a worst-case *quadratic complexity*. To improve on this, we need to improve the scheme by which we select replica pairs $(\mathtt{R}_1, \mathtt{R}_2) \in C_1 \times C_2$ that we use in cluster-sending steps. The straightforward manner to guarantee a worst-case *linear complexity* is by using a scheme that can select only up-to-$n = \max(\mathbf{n}_{C_1}, \mathbf{n}_{C_2})$ distinct pairs $(\mathtt{R}_1, \mathtt{R}_2) \in C_1 \times C_2$. To select $n$ replica pairs from $C_1 \times C_2$, we will proceed in two steps.

1. We generate list $S_1$ of $n$ replicas taken from $C_1$ and list $S_2$ of $n$ replicas taken from $C_2$.

2. Then, we choose permutations $P_1 \in \mathsf{perms}(S_1)$ and $P_2 \in \mathsf{perms}(S_2)$ fully at random, and interpret each pair $(P_1[i], P_2[i])$. $0 \leq i < n$, as one of the chosen replica pairs.

We use the first step to deal with any differences in the sizes of $C_1$ and $C_2$, and we use the second step to introduce sufficient

randomness in our protocol to yield an low expected-case message complexity.

Next, we introduce some notations to simplify reasoning about the above list-based scheme. If $R$ is a set of replicas, then $\mathsf{list}(R)$ is the list consisting of the replicas in $R$ placed in a predetermined order (e.g., on increasing replica identifier). If $S$ is a list of replicas, then we write $\mathsf{f}(S)$ to denote the faulty replicas in $S$ and $\mathsf{nf}(S)$ to denote the non-faulty replicas in $S$, and we write $\mathbf{n}_S = |S|$, $\mathbf{f}_S = |\{i \mid (0 \leq i < \mathbf{n}_S) \wedge S[i] \in \mathsf{f}(S)\}|$, and $\mathbf{nf}_S = \mathbf{n}_S - \mathbf{f}_S$ to denote the number of positions in $S$ with replicas, faulty replicas, and non-faulty replicas, respectively. If $(P_1, P_2)$ is a pair of equal-length lists of $n = |P_1| = |P_2|$ replicas, then we say that the $i$-th position is a *faulty position* if either $P_1[i] \in \mathsf{f}(P_1)$ or $P_2[i] \in \mathsf{f}(P_2)$. We write $\|P_1; P_2\|_{\mathbf{f}}$ to denote the number of *faulty positions* in $(P_1, P_2)$. As faulty positions can only be constructed out of the $\mathbf{f}_{P_1}$ faulty replicas in $P_1$ and the $\mathbf{f}_{P_2}$ faulty replicas in $P_2$, we must have $\max(\mathbf{f}_{P_1}, \mathbf{f}_{P_2}) \leq \|P_1; P_2\|_{\mathbf{f}} \leq \min(n, \mathbf{f}_{P_1} + \mathbf{f}_{P_2})$.

*Example* 5.1. Consider clusters $C_1, C_2$ with

$$S_1 = \mathsf{list}(C_1) = [R_{1,1}, \ldots, R_{1,5}], \quad \mathsf{f}(C_1) = \{R_{1,1}, R_{1,2}\};$$
$$S_2 = \mathsf{list}(C_2) = [R_{2,1}, \ldots, R_{2,5}], \quad \mathsf{f}(C_2) = \{R_{2,1}, R_{2,2}\}.$$

The set $\mathsf{perms}(S_1) \times \mathsf{perms}(S_2)$ contains $5!^2 = 14400$ list pairs. Now, consider the list pairs $(P_1, P_2), (Q_1, Q_2),$ $(R_1, R_2) \in \mathsf{perms}(S_1) \times \mathsf{perms}(S_2)$ with

$$P_1[R_{1,1}, R_{1,5}, R_{1,2}, R_{1,4}, R_{1,3}],$$
$$P_2[R_{2,1}, R_{2,3}, R_{2,2}, R_{2,5}, R_{2,4}];$$
$$Q_1[R_{1,1}, R_{1,3}, R_{1,5}, R_{1,4}, R_{1,2}],$$
$$Q_2[R_{2,5}, R_{2,4}, R_{2,3}, R_{2,2}, R_{2,1}];$$
$$R_1[R_{1,5}, R_{1,4}, R_{1,3}, R_{1,2}, R_{1,1}],$$
$$R_2[R_{2,1}, R_{2,2}, R_{2,3}, R_{2,4}, R_{2,5}].$$

We have underlined the faulty replicas in each list, and we have $\|P_1; P_2\|_{\mathbf{f}} = 2 = \mathbf{f}_{S_1} = \mathbf{f}_{S_2}$, $\|Q_1; Q_2\|_{\mathbf{f}} = 3$, and $\|R_1; R_2\|_{\mathbf{f}} = 4 = \mathbf{f}_{S_1} + \mathbf{f}_{S_2}$.

In the following, we will use a *list-pair function* $\Phi$ to compute the initial list-pair $(S_1, S_2)$ of $n$ replicas taken from $C_1$ and $C_2$, respectively. We build a cluster-sending protocol that uses $\Phi$ to compute $S_1$ and $S_2$, uses randomization to choose $n$ replica pairs from $S_1 \times S_2$, and, finally, performs cluster-sending steps using only these $n$ replica pairs. The pseudo-code of the resultant *Synchronous Probabilistic Linear Cluster-Sending protocol* CSPL can be found in Figure 5. Next, we prove that CSPL is correct and has a worst-case linear message complexity:

**Proposition 5.1.** *Let* $C_1, C_2$ *be disjoint clusters and let* $\Phi$ *be a list-pair function with* $(S_1, S_2) := \Phi(C_1, C_2)$ *and* $n = \mathbf{n}_{S_1} = \mathbf{n}_{S_2}$. *If communication is synchronous and* $n > \mathbf{f}_{S_1} + \mathbf{f}_{S_2}$, *then* CSPL$(C_1, C_2, v, \Phi)$ *results in cluster-sending* $v$ *from* $C_1$ *to* $C_2$.

**Protocol** CSPL$(C_1, C_2, v, \Phi)$**:**

1: Use *local consensus* on $v$ and construct $\langle\texttt{send}: v, C_2\rangle_{C_1}$.
2: {*Each replica in* $\mathsf{nf}(C_1)$ *decides* AGREE *on* $v$.}
3: Let $(S_1, S_2) := \Phi(C_1, C_2)$.
4: Choose $(P_1, P_2) \in \mathsf{perms}(S_1) \times \mathsf{perms}(S_2)$ fully at random.
5: $i := 0$.
6: **repeat**
7:     CS-STEP$(P_1[i], P_2[i], v)$
8:     Wait *three* global pulses.
9:     $i := i + 1$.
10: **until** $C_1$ reaches consensus on $\langle\texttt{proof}: \langle\texttt{send}: v, C_2\rangle_{C_1}\rangle_{C_2}$.

Figure 5: The Synchronous Probabilistic Linear Cluster-Sending protocol CSPL$(C_1, C_2, v, \Phi)$ that cluster-sends a value $v$ from $C_1$ to $C_2$ using list-pair function $\Phi$.

*The execution performs two local consensus steps in* $C_1$, *one local consensus step in* $C_2$, *and makes worst-case* $\mathbf{f}_{S_1} + \mathbf{f}_{S_2} + 1$ *cluster-sending steps.*

*Proof.* Due to Lines 1–2 of Figure 5, CSPL$(C_1, C_2, v, \Phi)$ establishes the pre-conditions for any execution of CS-STEP$(R_1, R_2, v)$ with $R_1 \in C_1$ and $R_2 \in C_2$. Now let $(P_1, P_2) \in \mathsf{perms}(S_1) \times \mathsf{perms}(S_2)$, as chosen at Line 4 of Figure 5. As $P_i$, $i \in \{1, 2\}$, is a permutation of $S_i$, we have $\mathbf{f}_{P_i} = \mathbf{f}_{S_i}$. Hence, we have $\|P_1; P_2\|_{\mathbf{f}} \leq \mathbf{f}_{S_1} + \mathbf{f}_{S_2}$ and there must exist a position $j$, $0 \leq j < n$, such that $(P_1[j], P_2[j]) \in \mathsf{nf}(C_1) \times \mathsf{nf}(C_2)$. Using the correctness of CS-STEP (Proposition 3.1), we conclude that CSPL$(C_1, C_2, v, \Phi)$ results in cluster-sending $v$ from $C_1$ to $C_2$ in at most $\mathbf{f}_{S_1} + \mathbf{f}_{S_2} + 1$ cluster-sending steps. Finally, the bounds on the number of consensus steps follow from an argument analogous to the one in the proof of Theorem 4.2. $\square$

Next, we proceed in two steps to arrive at practical instances of CSPL with expected constant message complexity. First, in Section 5.1, we study the probabilistic nature of CSPL. Then, in Section 5.2, we propose practical list-pair functions and show that these functions yield instances of CSPL with expected constant message complexity.

## 5.1 The Expected-Case Complexity of CSPL

As the first step to determine the expected-case complexity of CSPL, we solve the following abstract problem that captures the probabilistic argument at the core of the expected-case complexity of CSPL:

**Problem 5.2** (non-faulty position trials). Let $S_1$ and $S_2$ be lists of $|S_1| = |S_2| = n$ replicas. Choose permutations $(P_1, P_2) \in \mathsf{perms}(S_1) \times \mathsf{perms}(S_2)$ fully at random. Next, we inspect positions in $P_1$ and $P_2$ fully at random (with replacement). The *non-faulty position trials problem* asks how many positions one expects to inspect to find the first non-faulty position.

Let $S_1$ and $S_2$ be list of $|S_1| = |S_2| = n$ replicas. To answer the non-faulty position trials problem, we first look at the combinatorics of *faulty positions* in pairs $(P_1, P_2) \in$ perms$(S_1) \times$ perms$(S_2)$. Let $m_1 = \mathbf{f}_{S_1}$ and $m_2 = \mathbf{f}_{S_2}$. By $\mathbb{F}(n, m_1, m_2, k)$, we denote the number of distinct pairs $(P_1, P_2)$ one can construct that have exactly $k$ faulty positions, hence, with $\|P_1; P_2\|_{\mathbf{f}} = k$. As observed, we have $\max(m_1, m_2) \leq \|P_1; P_2\|_{\mathbf{f}} \leq \min(n, m_1 + m_2)$ for any pair $(P_1, P_2)$. Hence, we have $\mathbb{F}(n, m_1, m_2, k) = 0$ for all $k < \max(m_1, m_2)$ and $k > \min(n, m_1 + m_2)$.

Now consider the step-wise construction of any permutation $(P_1, P_2) \in$ perms$(S_1) \times$ perms$(S_2)$ with $k$ faulty positions. First, we choose $(P_1[0], P_2[0])$, the pair at position 0, after which we choose pairs for the remaining $n - 1$ positions. For $P_i[0]$, $i \in \{1, 2\}$, we can choose $n$ distinct replicas, of which $m_i$ are faulty. If we pick a non-faulty replica, then the remainder of $P_i$ is constructed out of $n - 1$ replicas, of which $m_i$ are faulty. Otherwise, the remainder of $P_i$ is constructed out of $n - 1$ replicas of which $m_i - 1$ are faulty. If, due to our choice of $(P_1[0], P_2[0])$, the first position is faulty, then only $k - 1$ out of the $n - 1$ remaining positions must be faulty. Otherwise, $k$ out of the $n - 1$ remaining positions must be faulty. Combining this analysis yields four types for the first pair $(P_1[0], P_2[0])$:

1. *A non-faulty pair* $(P_1[0], P_2[0]) \in$ nf$(P_1) \times$ nf$(P_2)$. We have $(n - m_1)(n - m_2)$ such pairs, and we have $\mathbb{F}(n - 1, m_1, m_2, k)$ different ways to construct the remainder of $P_1$ and $P_2$.

2. *A 1-faulty pair* $(P_1[0], P_2[0]) \in$ f$(P_1) \times$ nf$(P_2)$. We have $m_1(n - m_2)$ such pairs, and we have $\mathbb{F}(n - 1, m_1 - 1, m_2, k - 1)$ different ways to construct the remainder of $P_1$ and $P_2$.

3. *A 2-faulty pair* $(P_1[0], P_2[0]) \in$ nf$(P_1) \times$ f$(P_2)$. We have $(n - m_1)m_2$ such pairs, and we have $\mathbb{F}(n - 1, m_1, m_2 - 2, k - 1)$ different ways to construct the remainder of $P_1$ and $P_2$.

4. *A both-faulty pair* $(P_1[0], P_2[0]) \in$ f$(P_1) \times$ f$(P_2)$. We have $m_1 m_2$ such pairs, and we have $\mathbb{F}(n - 1, m_1 - 1, m_2 - 1, k - 1)$ different ways to construct the remainder of $P_1$ and $P_2$.

Hence, for all $k$, $\max(m_1, m_2) \leq k \leq \min(n, m_1 + m_2)$,

$\mathbb{F}(n, m_1, m_2, k)$ is recursively defined by:

$$\mathbb{F}(n, m_1, m_2, k) = (n - m_1)(n - m_2)\mathbb{F}(n - 1, m_1, m_2, k)$$
$$\text{(non-faulty pair)}$$
$$+ m_1(n - m_2)\mathbb{F}(n - 1, m_1 - 1, m_2, k - 1)$$
$$\text{(1-faulty pair)}$$
$$+ (n - m_1)m_2\mathbb{F}(n - 1, m_1, m_2 - 1, k - 1)$$
$$\text{(2-faulty pair)}$$
$$+ m_1 m_2\mathbb{F}(n - 1, m_1 - 1, m_2 - 1, k - 1),$$
$$\text{(both-faulty pair)}$$

and the base case for this recursion is $\mathbb{F}(0, 0, 0, 0) = 1$.

*Example* 5.3. Reconsider the list pairs $(P_1, P_2)$, $(Q_1, Q_2)$, and $(R_1, R_2)$ from Example 5.1. In $(P_1, P_2)$, we have both-faulty pairs at positions 0 and 2 and non-faulty pairs at positions 1, 3, and 4. In $(Q_1, Q_2)$, we have a 1-faulty pair at position 0, non-faulty pairs at positions 1 and 2, a 2-faulty pair at position 3, and a both-faulty pair at position 4. Finally, in $(R_1, R_2)$, we have 2-faulty pairs at positions 0 and 1, a non-faulty pair at position 2, and 1-faulty pairs at positions 3 and 4.

Using the combinatorics of faulty positions, we formalize an exact solution to the *non-faulty position trials problem*:

**Lemma 5.1.** *Let $S_1$ and $S_2$ be lists of $n = \mathbf{n}_{S_1} = \mathbf{n}_{S_2}$ replicas with $m_1 = \mathbf{f}_{S_1}$ and $m_2 = \mathbf{f}_{S_2}$. If $m_1 + m_2 < n$, then the non-faulty position trials problem $\mathbb{E}(n, m_1, m_2)$ has solution*

$$\frac{1}{n!^2}\left(\sum_{k=\max(m_1, m_2)}^{m_1 + m_2} \frac{n}{n - k}\mathbb{F}(n, m_1, m_2, k)\right).$$

*Proof.* We have $|\text{perms}(S_1)| = |\text{perms}(S_2)| = n!$. Consequently, we have $|\text{perms}(S_1) \times \text{perms}(S_2)| = n!^2$ and we have probability $1/(n!^2)$ to choose any pair $(P_1, P_2) \in$ perms$(S_1) \times$ perms$(S_2)$. Now consider such a pair $(P_1, P_2) \in$ perms$(S_1) \times$ perms$(S_2)$. As there are $\|P_1; P_2\|_{\mathbf{f}}$ faulty positions in $(P_1, P_2)$, we have probability $p(P_1, P_2) = (n - \|P_1; P_2\|_{\mathbf{f}})/n$ to inspect a non-faulty position. Notice that $\max(m_1, m_2) \leq \|P_1; P_2\|_{\mathbf{f}} \leq m_1 + m_2 < n$ and, hence, $0 < p(P_1, P_2) \leq 1$. Each of the inspected positions in $(P_1, P_2)$ is chosen fully at random. Hence, each inspection is a *Bernoulli trial* with probability of success $p(P_1, P_2)$, and we expect to inspect a first non-faulty position in the $p(P_1, P_2)^{-1} = n/(n - \|P_1; P_2\|_{\mathbf{f}})$-th attempt. We conclude that the non-faulty position trials problem $\mathbb{E}(n, m_1, m_2)$ has solution

$$\frac{1}{n!^2}\left(\sum_{(P_1, P_2) \in \text{perms}(S_1) \times \text{perms}(S_2)} \frac{n}{n - \|P_1; P_2\|_{\mathbf{f}}}\right).$$

Notice that there are $\mathbb{F}(n, m_1, m_2, k)$ distinct pairs $(P_1, P_2) \in$ perms$(S_1) \times$ perms$(S_2)$ with $\|P_1'; P_2'\|_{\mathbf{f}} = k$ for each $k$, $\max(m_1, m_2) \leq k \leq m_1 + m_2 < n$. Hence, in the above expression for $\mathbb{E}(n, m_1, m_2)$, we can group on these pairs $(P_1', P_2')$ to obtain the searched-for solution. $\square$

To further solve the non-faulty position trials problem, we work towards a *closed form* for $\mathbb{F}(n,m_1,m_2,k)$. Consider any pair $(P_1,P_2) \in \mathsf{perms}(S_1) \times \mathsf{perms}(S_2)$ with $\|P_1;P_2\|_{\mathsf{f}} = k$ obtained via the outlined step-wise construction. Let $b_1$ be the number of 1-*faulty pairs*, let $b_2$ be the number of 2-*faulty pairs*, and let $b_{1,2}$ be the number of *both-faulty pairs* in $(P_1,P_2)$. By construction, we must have $k = b_1 + b_2 + b_{1,2}$, $m_1 = b_1 + b_{1,2}$, and $m_2 = b_2 + b_{1,2}$ and by rearranging terms, we can derive

$$b_{1,2} = (m_1 + m_2) - k, \quad b_1 = k - m_2, \quad b_2 = k - m_1.$$

*Example* 5.4. Consider

$$S_1 = [\mathrm{R}_{1,1}, \dots, \mathrm{R}_{1,5}], \qquad \mathsf{f}(S_1) = \{\mathrm{R}_{1,1}, \mathrm{R}_{1,2}, \mathrm{R}_{1,3}\};$$
$$S_2 = [\mathrm{R}_{2,1}, \dots, \mathrm{R}_{2,5}], \qquad \mathsf{f}(S_2) = \{\mathrm{R}_{2,1}\}.$$

Hence, we have $n = 5$, $m_1 = \mathbf{f}_{S_1} = 3$, and $m_2 = \mathbf{f}_{S_2} = 1$. If we want to create a pair $(P_1,P_2) \in \mathsf{perms}(S_1) \times \mathsf{perms}(S_2)$ with $k = \|P_1;P_2\|_{\mathsf{f}} = 3$ faulty positions, then $(P_1,P_2)$ must have two non-faulty pairs, two 1-faulty pairs, no 2-faulty pairs, and one both-faulty pair. Hence, we have $n - k = 2$, $b_1 = 2$, $b_2 = 0$, and $b_{1,2} = 1$.

The above analysis only depends on the choice of $m_1$, $m_2$, and $k$, and not on our choice of $(P_1,P_2)$. Next, we use this analysis to express $\mathbb{F}(n,m_1,m_2,k)$ in terms of the number of distinct ways in which one can *construct*

(A) lists of $b_1$ 1-faulty pairs out of faulty replicas from $S_1$ and non-faulty replicas from $S_2$,

(B) lists of $b_2$ 2-faulty pairs out of non-faulty replicas from $S_1$ and faulty replicas from $S_2$,

(C) lists of $b_{1,2}$ both-faulty pairs out of the remaining faulty replicas in $S_1$ and $S_2$ that are not used in the previous two cases, and

(D) lists of $n - k$ non-faulty pairs out of the remaining (non-faulty) replicas in $S_1$ and $S_2$ that are not used in the previous three cases;

and in terms of the number of distinct ways one can *merge* these lists. As the first step, we look at how many distinct ways we can merge two lists together:

**Lemma 5.2.** *For any two disjoint lists $S$ and $T$ with $|S| = v$ and $|T| = w$, there exist $\mathbb{M}(v,w) = (v+w)!/(v!w!)$ distinct lists $L$ with $L|_S = S$ and $L|_T = T$, in which $L|_M$, $M \in \{S,T\}$, is the list obtained from $L$ by only keeping the values that also appear in list $M$.*

Next, we look at the number of distinct ways in which one can construct lists of type A, B, C, and D. Consider the construction of a list of type A. We can choose $\binom{m_1}{b_1}$ distinct sets of $b_1$ faulty replicas from $S_1$ and we can choose $\binom{n-m_2}{b_1}$ distinct sets of $b_1$ non-faulty replicas from $S_2$. As we

can order the chosen values from $S_1$ and $S_2$ in $b_1!$ distinct ways, we can construct $b_1!^2 \binom{m_1}{b_1} \binom{n-m_2}{b_1}$ distinct lists of type A. Likewise, we can construct $b_2!^2 \binom{n-m_1}{b_2} \binom{m_2}{b_2}$ distinct lists of type B.

*Example* 5.5. We continue from the setting of Example 5.4: we want to create a pair $(P_1,P_2) \in \mathsf{perms}(S_1) \times \mathsf{perms}(S_2)$ with $k = \|P_1;P_2\|_{\mathsf{f}} = 3$ faulty positions. To create $(P_1,P_2)$, we need to create $b_1 = 2$ pairs that are 1-faulty. We have $\binom{m_1}{b_1} = \binom{3}{2} = 3$ sets of two faulty replicas in $S_1$ that we can choose, namely the sets $\{\mathrm{R}_{1,1}, \mathrm{R}_{1,2}\}$, $\{\mathrm{R}_{1,1}, \mathrm{R}_{1,3}\}$, and $\{\mathrm{R}_{1,2}, \mathrm{R}_{1,3}\}$. Likewise, we have $\binom{n-m_2}{b_1} = \binom{4}{2} = 6$ sets of two non-faulty replicas in $S_2$ that we can choose. Assume we choose $T_1 = \{\mathrm{R}_{1,1}, \mathrm{R}_{1,3}\}$ from $S_1$ and $T_2 = \{\mathrm{R}_{2,4}, \mathrm{R}_{2,5}\}$ from $S_2$. The two replicas in $T_1$ can be ordered in $\mathbf{n}_{T_1}! = 2! = 2$ ways, namely $[\mathrm{R}_{1,1}, \mathrm{R}_{1,3}]$ and $[\mathrm{R}_{1,3}, \mathrm{R}_{1,1}]$. Likewise, the two replicas in $T_2$ can be ordered in $\mathbf{n}_{T_2}! = 2! = 2$ ways. Hence, we can construct $2 \cdot 2 = 4$ distinct lists of type A out of this single choice for $T_1$ and $T_2$, and the sequences $S_1$ and $S_2$ provide us with $\binom{m_1}{b_1} \binom{n-m_2}{b_1} = 18$ distinct choices for $T_1$ and $T_2$. We conclude that we can construct 72 distinct lists of type A from $S_1$ and $S_2$.

By construction, lists of type A and type B cannot utilize the same replicas from $S_1$ or $S_2$. After choosing $b_1 + b_2$ replicas in $S_1$ and $S_2$ for the construction of lists of type A and B, the remaining $b_{1,2}$ faulty replicas in $S_1$ and $S_2$ are all used for constructing lists of type C. As we can order these remaining values from $S_1$ and $S_2$ in $b_{1,2}!$ distinct ways, we can construct $b_{1,2}!^2$ distinct lists of type C (per choice of lists of type A and B). Likewise, the remaining $n - k$ non-faulty replicas in $S_1$ and $S_2$ are all used for constructing lists of type D, and we can construct $(n-k)!^2$ distinct lists of type D (per choice of lists of type A and B).

As the final steps, we merge lists of type A and B into lists of type AB. We can do so in $\mathbb{M}(b_1,b_2)$ ways and the resultant lists have size $b_1 + b_2$. Next, we merge lists of type AB and C into lists of type ABC. We can do so in $\mathbb{M}(b_1+b_2,b_{1,2})$ ways and the resultant lists have size $k$. Finally, we merge list of type ABC and D together, which we can do in $\mathbb{M}(k,n-k)$ ways. From this construction, we derive that $\mathbb{F}(n,m_1,m_2,k)$ is equivalent to

$$b_1!^2 \binom{m_1}{b_1} \binom{n-m_2}{b_1} b_2!^2 \binom{n-m_1}{b_2} \binom{m_2}{b_2} \cdot$$
$$\mathbb{M}(b_1,b_2) b_{1,2}!^2 \mathbb{M}(b_1+b_2,b_{1,2})(n-k)!^2 \mathbb{M}(k,n-k),$$

which can be simplified to the following:

**Lemma 5.3.** *Let $\max(m_1,m_2) \le k \le \min(n,m_1+m_2)$ and let $b_1 = k - m_2$, $b_2 = k - m_1$, and $b_{1,2} = (m_1 + m_2) - k$. We have*

$$\mathbb{F}(n,m_1,m_2,k) = \frac{m_1!m_2!(n-m_1)!(n-m_2)n!}{b_1!b_2!b_{1,2}!(n-k)!}.$$

We combine Lemma 5.1 and Lemma 5.3 to conclude

**Proposition 5.2.** *Let $S_1$ and $S_2$ be lists of $n = \mathbf{n}_{S_1} = \mathbf{n}_{S_2}$ replicas with $m_1 = \mathbf{f}_{S_1}$, $m_2 = \mathbf{f}_{S_2}$, $b_1 = k - m_2$, $b_2 = k - m_1$, and $b_{1,2} = (m_1 + m_2) - k$. If $m_1 + m_2 < n$, then the non-faulty position trials problem $\mathbb{E}(n, m_1, m_2)$ has solution*

$$\frac{1}{n!^2}\left(\sum_{k=\max(m_1,m_2)}^{m_1+m_2} \frac{n}{n-k} \frac{m_1!m_2!(n-m_1)!(n-m_2)!n!}{b_1!b_2!b_{1,2}!(n-k)!}\right).$$

Finally, we use Proposition 5.2 to derive

**Proposition 5.3.** *Let $C_1, C_2$ be disjoint clusters and let $\Phi$ be a list-pair function with $(S_1, S_2) := \Phi(C_1, C_2)$ and $n = \mathbf{n}_{S_1} = \mathbf{n}_{S_2}$. If communication is synchronous and $\mathbf{f}_{S_1} + \mathbf{f}_{S_2} < n$, then the expected number of cluster-sending steps performed by* $\mathrm{CSPL}(C_1, C_2, v, \Phi)$ *is less than $\mathbb{E}(n, \mathbf{f}_{S_1}, \mathbf{f}_{S_2})$.*

*Proof.* Let $(P_1, P_2) \in \mathrm{perms}(S_1) \times \mathrm{perms}(S_2)$. We notice that CSPL inspects positions in $P_1$ and $P_2$ in a different way than the non-faulty trials problem: at Line 7 of Figure 5, positions are inspected one-by-one in a predetermined order and not fully at random (with replacement). Next, we will argue that $\mathbb{E}(n, \mathbf{f}_{S_1}, \mathbf{f}_{S_2})$ provides an upper bound on the expected number of cluster-sending steps regardless of these differences. Without loss of generality, we assume that $S_1$ and $S_2$ each have $n$ distinct replicas. Consequently, the pair $(P_1, P_2)$ represents a set $R$ of $n$ distinct replica pairs taken from $C_1 \times C_2$. We notice that each of the $n!$ permutations of $R$ is represented by a single pair $(P_1', P_2') \in \mathrm{perms}(S_1) \times \mathrm{perms}(S_2)$.

Now consider the selection of positions in $(P_1, P_2)$ fully at random, but without replacement. This process will yield a list $[j_0, \dots, j_{n-1}] \in \mathrm{perms}([0, \dots, n-1])$ of positions fully at random. Let $Q_i = [P_i[j_0], \dots, P_i[j_{n-1}]]$, $i \in \{1, 2\}$. We notice that the pair $(Q_1, Q_2)$ also represents $R$ and we have $(Q_1, Q_2) \in \mathrm{perms}(S_1) \times \mathrm{perms}(S_2)$. Hence, by choosing a pair $(P_1, P_2) \in \mathrm{perms}(S_1) \times \mathrm{perms}(S_2)$, we choose set $R$ fully at random and, at the same time, we choose the order in which replica pairs in $R$ are inspected fully at random.

Finally, we note that CSPL inspects positions without replacement. As the number of expected positions inspected in the non-faulty position trials problem decreases if we choose positions without replacement, we have proven that $\mathbb{E}(n, \mathbf{f}_{S_1}, \mathbf{f}_{S_2})$ is an upper bound on the expected number of cluster-sending steps. □

## 5.2 Practical Instances of CSPL

As the last step in providing practical instances of CSPL, we need to provide practical list-pair functions to be used in conjunction with CSPL. We provide two such functions that address most practical environments. Let $C_1, C_2$ be disjoint clusters, let $n_{\min} = \min(\mathbf{n}_{C_1}, \mathbf{n}_{C_2})$, and let $n_{\max} = \max(\mathbf{n}_{C_1}, \mathbf{n}_{C_2})$. We provide list-pair functions

$$\Phi_{\min}(C_1, C_2) \mapsto (\mathrm{list}(C_1)^{:n_{\min}}, \mathrm{list}(C_2)^{:n_{\min}}),$$
$$\Phi_{\max}(C_1, C_2) \mapsto (\mathrm{list}(C_2)^{:n_{\max}}, \mathrm{list}(C_2)^{:n_{\max}}),$$

in which $L^{:n}$ denotes the first $n$ values in the list obtained by repeating list $L$. Next, we illustrate usage of these functions:

*Example* 5.6. Consider clusters $C_1, C_2$ with

$$S_1 = \mathrm{list}(C_1) = [\mathrm{R}_{1,1}, \dots, \mathrm{R}_{1,9}];$$
$$S_2 = \mathrm{list}(C_2) = [\mathrm{R}_{2,1}, \dots, \mathrm{R}_{2,4}].$$

We have

$$\Phi_{\min}(C_1, C_2) = ([\mathrm{R}_{1,1}, \dots, \mathrm{R}_{1,4}], \mathrm{list}(C_2));$$
$$\Phi_{\max}(C_1, C_2) = (\mathrm{list}(C_1), [\mathrm{R}_{2,1}, \dots, \mathrm{R}_{2,4}, \mathrm{R}_{2,1}, \dots, \mathrm{R}_{2,4}, \mathrm{R}_{2,1}]).$$

Next, we combine $\Phi_{\min}$ and $\Phi_{\max}$ with CSPL, show that in practical environments $\Phi_{\min}$ and $\Phi_{\max}$ satisfy the requirements put on list-pair functions in Proposition 5.1 to guarantee termination and cluster-sending, and use these results to determine the expected constant complexity of the resulting instances of CSPL.

**Theorem 5.7.** *Let $C_1, C_2$ be disjoint clusters with synchronous communication.*

1. *If $n = \min(\mathbf{n}_{C_1}, \mathbf{n}_{C_2}) > 2\max(\mathbf{f}_{C_1}, \mathbf{f}_{C_2})$, then the expected number of cluster-sending steps performed by $\mathrm{CSPL}(C_1, C_2, v, \Phi_{\min})$ is upper bounded by 4. For every $(S_1, S_2) := \Phi_{\min}(C_1, C_2)$, we have $n = \mathbf{n}_{S_1} = \mathbf{n}_{S_2}$, $n > 2\mathbf{f}_{S_1}$, $n > 2\mathbf{f}_{S_2}$, and $n > \mathbf{f}_{S_1} + \mathbf{f}_{S_2}$*

2. *If $n = \min(\mathbf{n}_{C_1}, \mathbf{n}_{C_2}) > 3\max(\mathbf{f}_{C_1}, \mathbf{f}_{C_2})$, then the expected number of cluster-sending steps performed by $\mathrm{CSPL}(C_1, C_2, v, \Phi_{\min})$ is upper bounded by $2\frac{1}{4}$. For every $(S_1, S_2) := \Phi_{\min}(C_1, C_2)$, we have $n = \mathbf{n}_{S_1} = \mathbf{n}_{S_2}$, $n > 3\mathbf{f}_{S_1}$, $n > 3\mathbf{f}_{S_2}$, and $n > \mathbf{f}_{S_1} + \mathbf{f}_{S_2}$.*

3. *If $\mathbf{n}_{C_1} > 3\mathbf{f}_{C_1}$ and $\mathbf{n}_{C_2} > 3\mathbf{f}_{C_2}$, then the expected number of cluster-sending steps performed by $\mathrm{CSPL}(C_1, C_2, v, \Phi_{\max})$ is upper bounded by 3. For every $(S_1, S_2) := \Phi_{\max}(C_1, C_2)$, we have $n = \mathbf{n}_{S_1} = \mathbf{n}_{S_2} = \max(\mathbf{n}_{C_1}, \mathbf{n}_{C_2}) > \mathbf{f}_{S_1} + \mathbf{f}_{S_2}$ and either we have $\mathbf{n}_{C_1} \geq \mathbf{n}_{C_2}$, $n > 3\mathbf{f}_{S_1}$, and $n > 2\mathbf{f}_{S_2}$; or we have $\mathbf{n}_{C_2} \geq \mathbf{n}_{C_1}$, $n > 2\mathbf{f}_{S_1}$, and $n > 3\mathbf{f}_{S_2}$.*

*Each of these instance of CSPL results in cluster-sending $v$ from $C_1$ to $C_2$.*

*Proof.* First, we prove the properties of $\Phi_{\min}$ and $\Phi_{\max}$ claimed in the three statements of the theorem. In the first and second statement of the theorem, we have $\min(\mathbf{n}_{C_1}, \mathbf{n}_{C_2}) > c\max(\mathbf{f}_{C_1}, \mathbf{f}_{C_2})$, $c \in \{2, 3\}$. Let $(S_1, S_2) := \Phi_{\min}(C_1, C_2)$ and $n = \mathbf{n}_{S_1} = \mathbf{n}_{S_2}$. By definition of $\Phi_{\min}$, we have $n = \min(\mathbf{n}_{C_1}, \mathbf{n}_{C_2})$, in which case $S_i$, $i \in \{1, 2\}$, holds $n$ distinct replicas from $C_i$. Hence, we have $\mathbf{f}_{C_i} \geq \mathbf{f}_{S_i}$ and, as $n > c\max(\mathbf{f}_{C_1}, \mathbf{f}_{C_2}) \geq c\mathbf{f}_{C_i}$, also $n > c\mathbf{f}_{S_i}$. Finally, as $n > 2\mathbf{f}_{S_1}$ and $n > 2\mathbf{f}_{S_2}$, also $2n > 2\mathbf{f}_{S_1} + 2\mathbf{f}_{S_2}$ and $n > \mathbf{f}_{S_1} + \mathbf{f}_{S_2}$ holds.

In the last statement of the theorem, we have $\mathbf{n}_{C_1} > 3\mathbf{f}_{C_1}$ and $\mathbf{n}_{C_2} > 3\mathbf{f}_{C_2}$. Without loss of generality, we assume $\mathbf{n}_{C_1} \geq$

$\mathbf{n}_{C_2}$. Let $(S_1, S_2) := \Phi_{\max}(C_1, C_2)$ and $n = \mathbf{n}_{S_1} = \mathbf{n}_{S_2}$. By definition of $\Phi_{\max}$, we have $n = \max(\mathbf{n}_{C_1}, \mathbf{n}_{C_2}) = \mathbf{n}_{C_1}$. As $n = \mathbf{n}_{C_1}$, we have $S_1 = \mathrm{list}(C_1)$. Consequently, we also have $\mathbf{f}_{S_1} = \mathbf{f}_{C_1}$ and, hence, $\mathbf{n}_{S_1} > 3\mathbf{f}_{C_1}$. Next, we will show that $\mathbf{n}_{S_2} > 2\mathbf{f}_{S_2}$. Let $q = \mathbf{n}_{C_1} \operatorname{div} \mathbf{n}_{C_2}$ and $r = \mathbf{n}_{C_1} \operatorname{mod} \mathbf{n}_{C_2}$. We note that $\mathrm{list}(C_2)^{:n}$ contains $q$ full copies of $\mathrm{list}(C_2)$ and one partial copy of $\mathrm{list}(C_2)$. Let $T \subset C_2$ be the set of replicas in this partial copy. By construction, we have $\mathbf{n}_{S_2} = q\mathbf{n}_{C_2} + r > q3\mathbf{f}_{C_2} + \mathbf{f}_T + \mathbf{n}\mathbf{f}_T$ and $\mathbf{f}_{S_2} = q\mathbf{f}_{C_2} + \mathbf{f}_T$ with $\mathbf{f}_T \le \min(\mathbf{f}_{C_2}, r)$. As $q > 1$ and $\mathbf{f}_{C_2} \ge \mathbf{f}_T$, we have $q\mathbf{f}_{C_2} \ge \mathbf{f}_{C_2} \ge \mathbf{f}_T$. Hence, $\mathbf{n}_{S_2} > 3q\mathbf{f}_{C_2} + \mathbf{f}_T + \mathbf{n}\mathbf{f}_T > 2q\mathbf{f}_{C_2} + \mathbf{f}_{C_2} + \mathbf{f}_T + \mathbf{n}\mathbf{f}_T \ge 2(q\mathbf{f}_{C_2} + \mathbf{f}_T) + \mathbf{n}\mathbf{f}_T \ge 2\mathbf{f}_{S_2}$. Finally, as $n > 3\mathbf{f}_{S_1}$ and $n > 2\mathbf{f}_{S_2}$, also $2n > 3\mathbf{f}_{S_1} + 2\mathbf{f}_{S_2}$ and $n > \mathbf{f}_{S_1} + \mathbf{f}_{S_2}$ holds.

Now, we prove the upper bounds on the expected number of cluster-sending steps for $\mathrm{CSPL}(C_1, C_2, v, \Phi_{\min})$ with $\min(\mathbf{n}_{C_1}, \mathbf{n}_{C_2}) > 2\max(\mathbf{f}_{C_1}, \mathbf{f}_{C_2})$. By Proposition 5.3, the expected number of cluster-sending steps is upper bounded by $\mathbb{E}(n, \mathbf{f}_{S_1}, \mathbf{f}_{S_2})$. In the worst case, we have $n = 2f + 1$ with $f = \mathbf{f}_{S_1} = \mathbf{f}_{S_2}$. Hence, the expected number of cluster-sending steps is upper bounded by $\mathbb{E}(2f + 1, f, f)$, $f \ge 0$. We claim that $\mathbb{E}(2f + 1, f, f)$ simplifies to $\mathbb{E}(2f + 1, f, f) = 4 - 2/(f + 1) - f!^2/(2f)!$. Hence, for all $S_1$ and $S_2$, we have $\mathbb{E}(n, \mathbf{f}_{S_1}, \mathbf{f}_{S_2}) < 4$. An analogous argument can be used to prove the other upper bounds. $\square$

Note that the third case of Theorem 5.7 corresponds with cluster-sending between arbitrary-sized resilient clusters that each operate using Byzantine fault-tolerant consensus protocols.

*Remark* 5.8. The upper bounds on the expected-case complexity of instances of CSPL presented in Theorem 5.7 match the upper bounds for CSP presented in Corollary 4.4. This does not imply that the expected-case complexity for these protocols is the same, however, as the probability distributions that yield these expected-case complexities are very different. To see this, consider a system in which all clusters have $n$ replicas of which $f$, $n = 2f + 1$, are faulty. Next, we denote the expected number of cluster-sending steps of protocol $P$ by $\mathbf{E}_P$, and we have

$$\mathbf{E}_{\mathrm{CSP}} = \frac{(2f + 1)^2}{(f + 1)^2} = 4 - \frac{4f + 3}{(f + 1)^2};$$

$$\mathbf{E}_{\mathrm{CSPL}} = \mathbb{E}(2f + 1, f, f) = 4 - \frac{2}{(f + 1)} - \frac{f!^2}{(2f)!}.$$

In Figure 6, we have illustrated this difference by plotting the expected-case complexity of CSP and CSPL for systems with equal-sized clusters. In practice, we see that the expected-case complexity for CSP is slightly lower than the expected-case complexity for CSPL.

# 6 Asynchronous Communication

In the previous sections, we introduced CSP, CSPP, and CSPL, three probabilistic cluster-sending protocols with ex-

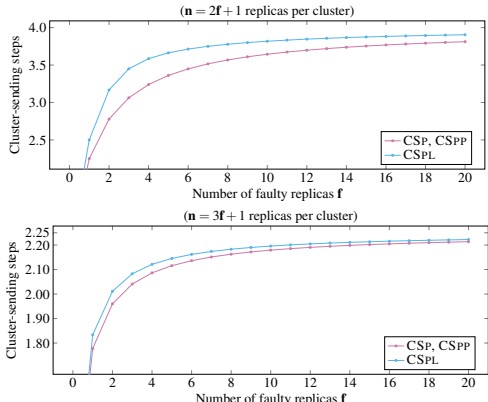

Figure 6: Comparison of the expected-case complexity of CSPL and CSP as a function of the number of faulty replicas.

pected constant message complexity. To simplify presentation, we have presented their design with respect to a synchronous environment. Next, we consider their usage in environments with asynchronous inter-cluster communication due to which messages can get arbitrary delayed, duplicated, or dropped.

We notice that the presented protocols *only* depend on synchronous communication to minimize communication: at the core of the correctness of CSP, CSPP, and CSPL is the cluster-sending step performed by CS-STEP, which does not make any assumptions on communication (Proposition 3.1). Consequently, CSP, CSPP, and CSPL can easily be generalized to operate in environments with asynchronous communication:

1. First, we observe that message duplication and out-of-order delivery have no impact on the cluster-sending step performed by CS-STEP. Hence, we do not need to take precautions against such asynchronous behavior.

2. If communication is asynchronous, but reliable (messages do not get lost, but can get duplicated, be delivered out-of-order, or get arbitrarily delayed), both CSPP and CSPL will be able to always perform cluster-sending in a finite number of steps. If communication becomes unreliable, however, messages sent between non-faulty replicas can get lost and all cluster-sending steps can fail. To deal with this, replicas in $C_1$ simply continue cluster-sending steps until a step succeeds (CSP) or rerun the protocol until a step succeeds (CSPP, and CSPL), which will eventually happen in an expected constant number steps whenever communication becomes reliable again.

3. If communication is asynchronous, then messages can get arbitrarily delayed. Fortunately, practical environments operate with large periods of reliable communication in which the majority of the messages arrive within some bounded delay unknown to $C_1$ and $C_2$. Hence, replicas in $C_1$ can simply assume some delay $\delta$. If this

delay is too short, then a cluster-sending step can *appear to fail* simply because the proof of receipt is still under way. In this case, cluster-sending will still be achieved when the proof of receipt arrives, but spurious cluster-sending steps can be initiated in the meantime. To reduce the number of such spurious cluster-sending steps, all non-faulty replicas in $C_1$ can use *exponential backup* to increase the message delay $\delta$ up-to-some reasonable upper bound (e.g., $100\,\text{s}$).

4. Finally, asynchronous environments often necessitate rather high assumptions on the message delay $\delta$. Consequently, the duration of a single failed cluster-sending step performed by CS-STEP will be high. Here, a trade-off can be made between *message complexity* and *duration* by starting several rounds of the cluster-sending step at once. E.g., when communication is sufficiently reliable, then all three protocols are expected to finish in four rounds or less, due to which starting four rounds initially will sharply reduce the duration of the protocol with only a constant increase in expected message complexity.

# 7  Performance evaluation

In the previous sections, we introduced probabilistic cluster-sending protocols with expected-case constant message complexity. To gain further insight in the performance attainable by these protocols, especially in environments with unreliable communication, we implemented these protocols in a simulated sharded resilient environment that allows us to control the faulty replicas and the message loss rates.[1] As a baseline of comparison, we also evaluated three cluster-sending protocols from the literature:

1. The *worst-case optimal cluster-sending protocol* PBS-CS of Hellings et al. [17] that can perform cluster-sending using only $\mathbf{f}_{C_1} + \mathbf{f}_{C_2} + 1$ messages, which is worst-case optimal. This protocol requires reliable communication.

2. The *broadcast-based cluster-sending protocol* of CHAINSPACE [1] that can perform cluster-sending using $\mathbf{n}_{C_1}\mathbf{n}_{C_2}$ messages. This protocol requires reliable communication.

3. The *global sharing protocol* of GEOBFT [15], an optimistic cluster-sending protocol that assumes that each cluster uses a primary-backup consensus protocol (e.g., PBFT [6]) and optimizes for the case in which the coordinating primary of $C_1$ is non-faulty. In this optimistic case, GEOBFT can perform cluster-sending using only $\mathbf{f}_{C_2} + 1$ messages. To deal with faulty primaries and

---
[1]The full implementation of this experiment is available at `anonymized`.

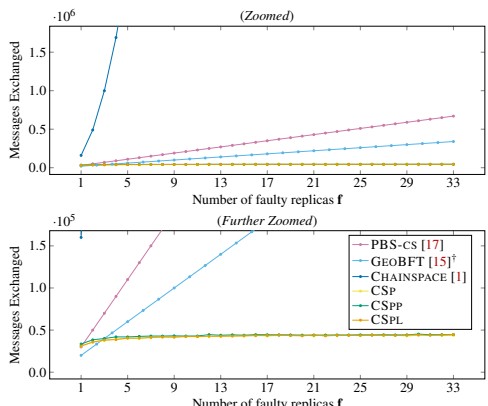

Figure 7: A comparison of the number of message exchange steps as a function of the number of faulty replicas in both clusters by our probabilistic cluster-sending protocols CSP, CSPP, and CSPL, and by three protocols from the literature. For each protocol, we measured the number of message exchange steps to send $10\,000$ values between two equally-sized clusters, each cluster having $n = 3\mathbf{f} + 1$ replicas. [†]The results for GEOBFT are a plot of the best-case optimistic phase of that protocol.

unreliable communication, GEOBFT employs a costly remote view-change protocol, however.

We refer to Figure 2 for an analytical comparison between these three cluster-sending protocols and our three probabilistic cluster-sending protocols.

In each experiment, we measured the number of messages exchanged in $10\,000$ runs of the cluster-sending protocol under consideration. In specific, in each run we measure the number of messages exchanged when sending a value $v$ from a cluster $C_1$ to a cluster $C_2$ with $\mathbf{n}_{C_1} = \mathbf{n}_{C_2} = 3\mathbf{f}_{C_1} + 1 = 3\mathbf{f}_{C_2} + 1$, and we aggregate this data over $10\,000$ runs. As we use equal-sized clusters, we have $\Phi_{\min}(C_1, C_2) = \Phi_{\max}(C_1, C_2)$ and, hence, we use a singe instance of CSPL.

Next, we detail the two experiments we performed and look at their results.

## 7.1  Performance of Cluster-Sending Protocols

In our first experiment, we measure the number of messages exchanged as a function of the number of faulty replicas. In this case, we assumed reliable communication, due to which we could include all six protocols. The results of this experiment can be found in Figure 7.

As is clear from the results, our probabilistic cluster-sending protocols are able to perform cluster-sending with only a constant number of messages exchanged. Furthermore, we see that the performance of our cluster-sending protocols matches the theoretical expected-case analysis in this paper and closely follows the expected performance illustrated in

Figure 6 (note that Figure 6 plots cluster-sending steps and each cluster-sending step involves the exchange of *two* messages between clusters).

As all other cluster-sending protocols have a linear (PBS-CS and GEOBFT) or quadratic (CHAINSPACE) message complexity, our probabilistic cluster-sending protocols outperform the other cluster-sending protocols. This is especially the case when dealing with bigger clusters, in which case the expected-case constant message complexity of our probabilistic cluster-sending protocols shows the biggest advantage. Only in the case of the smallest clusters can the other cluster-sending protocols outperform our probabilistic cluster-sending protocols, as PBS-CS, GEOBFT, and CHAINSPACE use reliable communication to their advantage to eliminate any acknowledgment messages send from the receiving cluster to the sending cluster. We believe that the slightly higher cost of our probabilistic cluster-sending protocols in these cases is justified, as our protocols can effectively deal with unreliable communication.

## 7.2 Message Loss

In our second experiment, we measure the number of messages exchanged as a function of the number of faulty replicas and as a function of the message loss (in percent) *between the two clusters*. We assume that communication within each cluster is reliable. In this case, we only included our probabilistic cluster-sending protocols as PBS-CS and CHAINSPACE both assume reliable communication and GEOBFT is only able to perform recovery via remote view-changes in periods of reliable communication. The results of this experiment can be found in Figure 8.

We note that with a message loss of *x*%, the probability $p(x)$ of a successful cluster-sending step is only $(1 - \frac{x}{100})^2$. E.g., $p(30\%) \approx 0.49$. As expected, the message complexity increases with an increase in message loss. Furthermore, the probabilistic cluster-sending protocols perform as expected (when taking into account the added cost to deal with message loss). These results further underline the practical benefits of each of the probabilistic cluster-sending protocols, especially for larger clusters: even in the case of high message loss rates, each of our probabilistic cluster-sending protocols are able to outperform the cluster-sending protocols PBS-CS, CHAINSPACE, and GEOBFT, which can only operate with reliable-communication.

## 8 Related Work

Although there is abundant literature on distributed systems and on consensus-based resilient systems (e.g., [2, 5, 8, 14, 16, 27, 31]), there is only limited work on communication *between* resilient systems [1, 15, 17]. In the previous section, we have already compared CSP, CSPP, and CSPL with the worst-case optimal cluster-sending protocols of Hellings et al. [17], the optimistic cluster-sending protocol of GEOBFT [15], and the broadcast-based cluster-sending protocols of CHAINSPACE [1]. Furthermore, we notice that *cluster-sending* can be solved using well-known Byzantine primitives such as consensus, interactive consistency, and Byzantine broadcasts [6, 9, 24]. These primitives are much more costly than cluster-sending protocols, however, and require huge amounts of communication between all involved replicas.

In parallel to the development of traditional resilient systems and permissioned blockchains, there has been promising work on sharding in permissionless blockchains such as BITCOIN [25] and ETHEREUM [32]. Examples include techniques for enabling reliable cross-chain coordination via sidechains, blockchain relays, atomic swaps, atomic commitment, and cross-chain deals [12, 13, 19, 21, 22, 33, 34]. Unfortunately, these techniques are deeply intertwined with the design goals of permissionless blockchains in mind (e.g., cryptocurrency-oriented), and are not readily applicable to traditional consensus-based Byzantine clusters.

## 9 Conclusion

In this paper, we presented probabilistic cluster-sending protocols that each provide highly-efficient solutions to the cluster-sending problem. In specific, our probabilistic cluster-sending protocols can facilitate communication between Byzantine fault-tolerant clusters with expected constant communication between clusters. For practical environments, our protocols can support worst-case linear communication between clusters, which is optimal, and deal with asynchronous and unreliable communication. The low practical cost of our cluster-sending protocols further enables the development and deployment of high-performance systems that are constructed out of Byzantine fault-tolerant clusters, e.g., fault-resilient geo-aware sharded data processing systems.

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

## A The proof of Lemma 5.2

To get the intuition behind the closed form of Lemma 5.2, we take a quick look at the combinatorics of *list-merging*. Notice that we can merge lists $S$ and $T$ together by either first taking an element from $S$ or first taking an element from $T$. This approach towards list-merging yields the following recursive solution to the list-merge problem:

$$\mathbb{M}(v,w) = \begin{cases} \mathbb{M}(v-1,w) + \mathbb{M}(v,w-1) & \text{if } v > 0 \text{ and } w > 0; \\ 1 & \text{if } v = 0 \text{ or } w = 0. \end{cases}$$

Consider lists $S$ and $T$ with $|S| = v$ and $|T| = w$ distinct values. We have $|\mathsf{perms}(S)| = v!$, $|\mathsf{perms}(T)| = w!$, and $|\mathsf{perms}(S \cup T)| = (v+w)!$. We observe that every list-merge of $(P_S, P_T) \in \mathsf{perms}(S) \times \mathsf{perms}(T)$ is a unique value in $\mathsf{perms}(S \cup T)$. Furthermore, every value in $\mathsf{perms}(S \cup T)$

can be constructed by such a list-merge. As we have $|\text{perms}(S) \times \text{perms}(T)| = v!w!$, we derive the closed form

$$\mathbb{M}(v,w) = \frac{(v+w)!}{(v!w!)}$$

of Lemma 5.2. Next, we formally prove this closed form.

*Proof.* We prove this by induction. First, the base cases $\mathbb{M}(0,w)$ and $\mathbb{M}(v,0)$. We have

$$\mathbb{M}(0,w) = \frac{(0+w)!}{0!w!} = \frac{w!}{w!} = 1;$$
$$\mathbb{M}(v,0) = \frac{(v+0)!}{v!0!} = \frac{v!}{v!} = 1.$$

Next, we assume that the statement of the lemma holds for all non-negative integers $v',w'$ with $0 \le v'+w' \le j$. Now consider non-negative integers $v,w$ with $v+w = j+1$. We assume that $v > 0$ and $w > 0$, as otherwise one of the base cases applies. Hence, we have

$$\mathbb{M}(v,w) = \mathbb{M}(v-1,w) + \mathbb{M}(v,w-1).$$

We apply the induction hypothesis on the terms $\mathbb{M}(v-1,w)$ and $\mathbb{M}(v,w-1)$ and obtain

$$\mathbb{M}(v,w) = \left(\frac{((v-1)+w)!}{(v-1)!w!}\right) + \left(\frac{(v+(w-1))!}{v!(w-1)!}\right).$$

Next, we apply $x = x(x-1)!$ and simplify the result to obtain

$$\mathbb{M}(v,w) = \left(\frac{v(v+w-1)!}{v!w!}\right) + \left(\frac{w(v+w-1)!}{v!w!}\right)$$
$$= \left(\frac{(v+w)(v+w-1)!}{v!w!}\right) = \frac{(v+w)!}{v!w!},$$

which completes the proof. □

## B  The proof of Lemma 5.3

Let $g$ be the expression

$$b_1!^2 \binom{m_1}{b_1}\binom{n-m_2}{b_1} b_2!^2 \binom{n-m_1}{b_2}\binom{m_2}{b_2} \cdot$$
$$\mathbb{M}(b_1,b_2)b_{1,2}!^2\mathbb{M}(b_1+b_2,b_{1,2})(n-k)!^2\mathbb{M}(k,n-k),$$

as stated right above Lemma 5.3. We will show that $g$ is equivalent to the closed form of $\mathbb{F}(n,m_1,m_2,k)$, as stated in Lemma 5.3.

*Proof.* We use the shorthands $\mathbf{T}_1 = \binom{m_1}{b_1}\binom{n-m_2}{b_1}$ and $\mathbf{T}_2 = \binom{n-m_1}{b_2}\binom{m_2}{b_2}$, and we have

$$g = b_1!^2\mathbf{T}_1 b_2!^2\mathbf{T}_2 \cdot$$
$$\mathbb{M}(b_1,b_2)b_{1,2}!^2\mathbb{M}(b_1+b_2,b_{1,2})(n-k)!^2\mathbb{M}(k,n-k).$$

We apply Lemma 5.2 on terms $\mathbb{M}(b_1,b_2)$, $\mathbb{M}(b_1+b_2,b_{1,2})$, and $\mathbb{M}(k,n-k)$, apply $k = b_1+b_2+b_{1,2}$, and simplify to derive

$$g = b_1!^2\mathbf{T}_1 b_2!^2\mathbf{T}_2 \cdot$$
$$\frac{(b_1+b_2)!}{b_1!b_2!}b_{1,2}!^2\frac{(b_1+b_2+b_{1,2})!}{(b_1+b_2)!b_{1,2}!}(n-k)!^2\frac{(k+n-k)!}{k!(n-k)!}$$
$$= b_1!\mathbf{T}_1 b_2!\mathbf{T}_2 b_{1,2}!(n-k)!n!.$$

Finally, we expand the binomial terms $\mathbf{T}_1$ and $\mathbf{T}_2$, apply $b_{1,2} = m_1-b_1 = m_2-b_2$ and $k = m_1+b_2 = m_2+b_1$, and simplify to derive

$$g = b_1!\frac{m_1!}{b_1!(m_1-b_1)!}\frac{(n-m_2)!}{b_1!(n-m_2-b_1)!} \cdot$$
$$b_2!\frac{(n-m_1)!}{b_2!(n-m_1-b_2)!}\frac{m_2!}{b_2!(m_2-b_2)!} \cdot$$
$$b_{1,2}!(n-k)!n!$$
$$= \frac{m_1!}{b_{1,2}!}\frac{(n-m_2)!}{b_1!(n-k)!}\frac{(n-m_1)!}{b_2!(n-k)!}\frac{m_2!}{b_{1,2}!}b_{1,2}!(n-k)!n!$$
$$= \frac{m_1!m_2!(n-m_1)!(n-m_2)!n!}{b_1!b_2!b_{1,2}!(n-k)!},$$

which completes the proof. □

## C  The proof of Lemma 5.3

*Proof.* We write $f(n,m_1,m_2,k)$ for the closed form in the statement of this lemma and we prove the statement of this lemma by induction. First, the base case $\mathbb{F}(0,0,0,0)$. In this case, we have $n = m_1 = m_2 = k = 0$ and, hence, $b_1 = b_2 = b_{1,2} = 0$, and we conclude $f(0,0,0,0) = 1 = \mathbb{F}(0,0,0,0)$.

Now assume $\mathbb{F}(n',m_1',m_2',k') = f(n',m_1',m_2',k')$ for all $n' < n$ and all $k'$ with $\max(m_1',m_2') \le k' \le \min(n',m_1'+m_2')$. Next, we prove $\mathbb{F}(n,m_1,m_2,k) = f(n,m_1,m_2,k)$ with $\max(m_1,m_2) \le k \le \min(n,m_1+m_2)$. We use the shorthand $\mathbb{G} = \mathbb{F}(n,m_1,m_2,k)$ and we have

$$\mathbb{G} = (n-m_1)(n-m_2)\mathbb{F}(n-1,m_1,m_2,k)$$
$$\text{(non-faulty pair)}$$
$$+ m_1(n-m_2)\mathbb{F}(n-1,m_1-1,m_2,k-1)$$
$$\text{(1-faulty pair)}$$
$$+ (n-m_1)m_2\mathbb{F}(n-1,m_1,m_2-1,k-1)$$
$$\text{(2-faulty pair)}$$
$$+ m_1 m_2\mathbb{F}(n-1,m_1-1,m_2-1,k-1).$$
$$\text{(both-faulty pair)}$$

Notice that if $n = k$, then the non-faulty pair case does not apply, as $\mathbb{F}(n-1,m_1,m_2,k) = 0$, and evaluates to zero. Likewise, if $b_1 = 0$, then the 1-faulty pair case does not apply, as $\mathbb{F}(n-1,m_1-1,m_2,k-1) = 0$, and evaluates to zero;

if $b_2 = 0$, then the 2-faulty pair case does not apply, as $\mathbb{F}(n-1, m_1, m_2-1, k-1) = 0$, and evaluates to zero; and, finally, if $b_{1,2} = 0$, then the both-faulty pair case does not apply, as $\mathbb{F}(n-1, m_1-1, m_2-1, k-1) = 0$, and evaluates to zero.

First, we consider the case in which $n > k$, $b_1 > 0$, $b_2 > 0$, and $b_{1,2} > 0$. Hence, each of the four cases apply and evaluate to non-zero values. We directly apply the induction hypothesis on $\mathbb{F}(n-1, m_1, m_2, k)$, $\mathbb{F}(n-1, m_1-1, m_2, k-1)$, $\mathbb{F}(n-1, m_1, m_2-1, k-1)$, and $\mathbb{F}(n-1, m_1-1, m_2-1, k-1)$, and obtain

$$
\begin{aligned}
\mathbb{G} = {} & (n-m_1)(n-m_2) \cdot \\
& \frac{m_1! m_2! (n-1-m_1)! (n-1-m_2)! (n-1)!}{b_1! b_2! b_{1,2}! (n-1-k)!} \\
& + m_1(n-m_2) \cdot \\
& \frac{(m_1-1)! m_2! (n-m_1)! (n-1-m_2)! (n-1)!}{(b_1-1)! b_2! b_{1,2}! (n-1-(k-1))!} \\
& + (n-m_1) m_2 \cdot \\
& \frac{m_1! (m_2-1)! (n-1-m_1)! (n-m_2)! (n-1)!}{b_1! (b_2-1)! b_{1,2}! (n-1-(k-1))!} \\
& + m_1 m_2 \cdot \\
& \frac{(m_1-1)! (m_2-1)! (n-m_1)! (n-m_2)! (n-1)!}{b_1! b_2! (b_{1,2}-1)! (n-1-(k-1))!}.
\end{aligned}
$$

We apply $x! = x(x-1)!$ and further simplify and obtain

$$
\begin{aligned}
\mathbb{G} = {} & \frac{m_1! m_2! (n-m_1)! (n-m_2)! (n-1)!}{b_1! b_2! b_{1,2}! (n-1-k)!} \\
& + \frac{m_1! m_2! (n-m_1)! (n-m_2)! (n-1)!}{(b_1-1)! b_2! b_{1,2}! (n-k)!} \\
& + \frac{m_1! m_2! (n-m_1)! (n-m_2)! (n-1)!}{b_1! (b_2-1)! b_{1,2}! (n-k)!} \\
& + \frac{m_1! m_2! (n-m_1)! (n-m_2)! (n-1)!}{b_1! b_2! (b_{1,2}-1)! (n-k)!} \\
= {} & (n-k) \frac{m_1! m_2! (n-m_1)! (n-m_2)! (n-1)!}{b_1! b_2! b_{1,2}! (n-k)!} \\
& + b_1 \frac{m_1! m_2! (n-m_1)! (n-m_2)! (n-1)!}{b_1! b_2! b_{1,2}! (n-k)!} \\
& + b_2 \frac{m_1! m_2! (n-m_1)! (n-m_2)! (n-1)!}{b_1! b_2! b_{1,2}! (n-k)!} \\
& + b_{1,2} \frac{m-1! m_2! (n-m_1)! (n-m_2)! (n-1)!}{b_1! b_2! b_{1,2}! (n-k)!}.
\end{aligned}
$$

We have $k = b_1 + b_2 + b_{1,2}$ and, hence, $n = (n-k) + b_1 + b_2 + b_{1,2}$ and we conclude

$$
\begin{aligned}
\mathbb{G} = {} & ((n-k) + b_1 + b_2 + b_{1,2}) \cdot \\
& \frac{m_1! m_2! (n-m_1)! (n-m_2)! (n-1)!}{b_1! b_2! b_{1,2}! (n-k)!} \\
= {} & n \frac{m_1! m_2! (n-m_1)! (n-m_2)! (n-1)!}{b_1! b_2! b_{1,2}! (n-k)!} \\
= {} & \frac{m_1! m_2! (n-m_1)! (n-m_2)! n!}{b_1! b_2! b_{1,2}! (n-k)!}.
\end{aligned}
$$

Next, in all other cases, we can repeat the above derivation while removing the terms corresponding to the cases that evaluate to 0. By doing so, we end up with the expression

$$
\mathbb{G} = \frac{((\sum_{t \in T} t) m_1! m_2! (n-m_1)! (n-m_2)! (n-1)!}{b_1! b_2! b_{1,2}! (n-k)!}.
$$

in which $T$ contains the term $(n-k)$ if $n > k$ (the non-faulty pair case applies), the term $b_1$ if $b_1 > 0$ (the 1-faulty case applies), the term $b_2$ if $b_2 > 0$ (the 2-faulty case applies), and the term $b_{1,2}$ if $b_{1,2} > 0$ (the both-faulty case applies). As each term $(n-k)$, $b_1$, $b_2$, and $b_{1,2}$ is in $T$ whenever the term is non-zero, we have $\sum_{t \in T} t = (n-k) + b_1 + b_2 + b_{1,2} = n$. Hence, we can repeat the steps of the above derivation in all cases, and complete the proof. □

## D    The Closed Form of $\mathbb{E}(2f+1, f, f)$

Here, we shall prove that

$$
\mathbb{E}(2f+1, f, f) = 4 - \frac{2}{(f+1)} - \frac{f!^2}{(2f)!}.
$$

*Proof.* By Proposition 5.2 and some simplifications, we have

$$
\begin{aligned}
\mathbb{E}(2f+1, f, f) = {} & \frac{1}{(2f+1)!^2} \cdot \\
& \left( \sum_{k=f}^{2f} \frac{2f+1}{2f+1-k} \frac{f!^2 (f+1)!^2 (2f+1)!}{(k-f)!^2 (2f-k)! (2f+1-k)!} \right).
\end{aligned}
$$

First, we apply $x! = x(x-1)!$, simplify, and obtain

$$
\begin{aligned}
\mathbb{E}(2f+1, f, f) = {} & \frac{f!^2 (2f+1)}{(2f+1)!} \cdot \\
& \left( \sum_{k=f}^{2f} \frac{(f+1)!^2}{(k-f)!^2 (2f+1-k)!^2} \right) \\
= {} & \frac{f!^2}{(2f)!} \left( \sum_{k=0}^{f} \frac{(f+1)!^2}{k!^2 (f+1-k)!^2} \right) \\
= {} & \frac{f!^2}{(2f)!} \left( \sum_{k=0}^{f} \binom{f+1}{k}^2 \right).
\end{aligned}
$$

Next, we apply $\binom{m}{n} = \binom{m}{m-n}$, extend the sum by one term, and obtain

$$\mathbb{E}(2f+1,f,f) = \frac{f!^2}{(2f)!} \cdot$$

$$\left( \left( \sum_{k=0}^{f+1} \binom{f+1}{k} \binom{f+1}{f+1-k} \right) - \binom{f+1}{f+1} \binom{f+1}{0} \right).$$

Then, we apply Vandermonde's Identity to eliminate the sum and obtain

$$\mathbb{E}(2f+1,f,f) = \frac{f!^2}{(2f)!} \left( \binom{2f+2}{f+1} - 1 \right).$$

Finally, we apply straightforward simplifications and obtain

$$\begin{aligned}
\mathbb{E}(2f+1,f,f) &= \frac{f!^2}{(2f)!} \frac{(2f+2)!}{(f+1)!(f+1)!} - \frac{f!^2}{(2f)!} \\
&= \frac{f!^2}{(2f)!} \frac{(2f)!(2f+1)(2f+2)}{f!^2(f+1)^2} - \frac{f!^2}{(2f)!} \\
&= \frac{(2f+1)(2f+2)}{(f+1)^2} - \frac{f!^2}{(2f)!} \\
&= \frac{(2f+2)^2}{(f+1)^2} - \frac{2f+2}{(f+1)^2} - \frac{f!^2}{(2f)!} \\
&= \frac{4(f+1)^2}{(f+1)^2} - \frac{2(f+1)}{(f+1)^2} - \frac{f!^2}{(2f)!} \\
&= 4 - \frac{2}{f+1} - \frac{f!^2}{(2f)!},
\end{aligned}$$

which completes the proof. $\qquad\square$

