# OpenReview forum: "Solution: Byzantine Cluster-Sending in Expected Constant Cost and Constant Time"
_JSYS/2022/May_Papers — Reject_

### Official Review · Reviewer_ZfrY · 2022-06-13
**The paper presents a solution for getting two Byzantine clusters to agree on a value exchanged between them. Their solution involves probabilistic selection of two "good" nodes on each cluster, and then getting those nodes to exchange and disperse the values within their own cluster. While I believe the problem is important in today's landscape and the protocol is sound, the contributions of the paper is limited, in the protocol design, the analysis, and the evaluation.**

**Decision:**

Strong reject: this paper has serious problems, fixing it would definitely take more than three months

**Review:**

The main strength of the paper, in my opinion, is bringing to attention a
problem that has not been considered extensively yet in the field: getting two
different Byzantine concensus clusters to exchange and agree on a value. The
problem setting is interesting: the bandwidth between clusters is indeed very
pricey (often the most expensive aspect of cloud computing today), and thus the
goal to minimize the inter-cluster messaging is important. The solution also
seems correct, and is likely simple enough to implement in practice.

Unfortunately, I have several issues with various aspects of the paper. First,
the solution itself is not very novel. The basic building block, CS-Step, is
essentially a two-phase commit protocol between two clusters. Then, the
extensions, CSp and CSpl, are just randomized versions of the basic algorithm
to select nodes, which is a common process in Byzantine consensus
and agreement algorithms. There may be some interesting insights in the paper,
but as presented, it was not clear.

If not for the novelty of the algorithm itself, I think the analysis of the
algorithm should be interesting and/or provide new insights.
The authors indeed carefully analyze a closed form solution of probability of non-faulty
position. This, for example, yields the bounds of $4$ expected steps and $\frac{9}{4}$ expected steps in cases 1 and 2 of
Theorem 5.7. This analysis, however, seems unnecessarily complex. When considering
$\Phi_{min}$ in the first case of Theorem 5.7,
the analysis is simply about the number of trials before selecting two good nodes
without replacement. This is upper bounded by the same experiment
but with replacement, since a failed node is removed when considering without replacement.
WLOG, assume $m_1 > m_2$. In this case, the probability of success is
\begin{align*}
  \Pr[\text{Selecting two good nodes without replacement}] & > \Pr[\text{Selecting two good nodes with replacement}] \\\\
  & > (1 - \frac{m_1}{n})(1 - \frac{m_2}{n}) \\\\
  & > (1 - \frac{1}{2})(1 - \frac{1}{2}) \\\\
  & = \frac{1}{4}
\end{align*}
The "with replacement" case becomes a simple Bernoulli trial with $p > \frac{1}{4}$,
which then has expected number of trial of < $\frac{1}{p}$. This provides
the same bound of 4 steps, with a much simpler analysis.
Similar analysis for case 2 of Theorem 5.7 as well, where $p$ is bounded below
by $(1-\frac{1}{3})(1-\frac{1}{3}) = \frac{4}{9}$, and thus expected number of trials
less than $\frac{9}{4}$, which again is the same bound as the analysis gives in the paper.
Similar analysis can be applied to $\Phi_{max}$ for the same result as case 3.
Would it be possible to get a tighter expected value than this loose analysis?

The evaluation also needs improvement. For instance, while the expected number of steps is important, the tail latency is also important.
For example, what is the 99% or 99.9% messaging latency
for various sizes of clusters and number of faulty nodes?
This might also be a place where an exact closed form analysis of the probability might be beneficial, as you can analyze the variance and other properties of the distribution more theoretically, not just empirically.
In general, the evaluation also
probably could benefit from showing more realistic results. E.g., running two small clusters
as shown in Figure 1 (like, one in Europe and one in US), and measuring end to end latency
to run the full protocol.


Minor comments:

- How does one perform "Choose replicas" in step 4 of CSp? This step also seem to require a Byzantine consensus, or a reliable (random) beacon. It's possible to bootstrap this with a random initial seed, and using a PRNG from that seed to locally determine the randomness for that round, but I believe it requires some care.
- The protocol doesn't seem to generalize that well to multiple clusters. As in, if a cluster needs to send values to $k$ clusters and all clusters need to be sure that all clusters saw the value, then the number of inter-cluster messages seems to scale quadratically in $k$ with naive extension of this scheme.
- Table of notations somewhere would help the reader, given the lengthy probability analysis with many terms.
- The graphs are difficult to read with various lines looking very similar to each other. The zooming doesn't help readability here.


**Expertise:**

Published in this area in the last 5 years

**Useful:**

no

---

### Official Review · Reviewer_8v6c · 2022-06-14
**A paper that formalizes the "cluster-sending" problem, proposes a solution and analyzes it. The problem is relevant and the solution intuitive, writing should be improved.**

**Decision:**

Weak accept: good paper with flaws that can be fixed in three months

**Review:**

In general, the cluster-sending problem, as described by the authors, is very relevant. Reliable communication between two fault-tolerant node clusters is becoming more and more important in today's sharded (and other) systems.

It is a bit questionable to me how practically relevant the chosen model of synchronous rounds is in practice, especially when the motivation seems to be blockchain systems and other large-scale geo-distributed systems, where communication in synchronous rounds is generally hard to achieve. Together with completely neglecting the intra-cluster overhead, this creates a very specific model whose relation to practice could have been better argumented. More than number of cross-cluster messages exchanged, it would be interesting to also study the expected / worst-case latency in of cluster-sending in a realistic system. Also, the authors clearly state that they consider a Byzantine model, but at the same time suggest Paxos, a CFT protocol, as an example of a protocol for intra-cluster agreement.

Although some parts of the paper make use of rather extensive formalism, the very definition of the cluster-sending problem is not very clear. Although Definition 2.1 conveys a reasonable intuition behind the problem, in itself it is rather vague. E.g., the terms RECEIVE, CONFIRM, and AGREE are not properly defined and even later on in the paper their exact definitions are not properly specified. I would suggest defining proper abstractions to model "local consensus" and communication between nodes / clusters and expressing the cluster-sending problem in terms of those.

That said, I still find the result useful, if nothing else, then at least as a stepping stone for studying the communication between groups of nodes. Analyzing the "number of attempts" (as I interpret the number of cluster-sending steps) required to transfer a message is useful in its own right.
The problem, as framed by the authors, seems to boil down to finding a pair of correct nodes, one from each cluster. The solutions seem rather straight-forward (which is a good thing) and the results intuitive.

I was a bit puzzled by the claims around optimality referring to [17], as it is a 3-page brief announcement that presents high-level intuitions about the cluster-sendig problem, but does not prove any lower bounds on its solution.

When talking about the complexity of the proposed algorithms, I did not fully understand how the number of "agreements" was counted, especially in the case where one of the selected nodes is faulty. In particular, is it possible for nodes to participate in an agreement protocol that fails due to a faulty proposer? Also, at the beginning of page 5, I did not properly understand the implicit / free consensus. Indeed, nodes in a cluster first need to agree on some operation that triggers the cluster-sending, and thus agreeing on a message can be implicit. However, I do not see how, at the same time, agreeing on the reception of a value can also be considered "free". In such a case, it seems to me, the whole cluster-sending could be implicit and no algorithm would be necessary in the first place.

I like the progressive approach to explaining the protocols and their analyses, starting simple and gradually moving to wards the more complex ones. Some of the proofs / analyses were rather hard to read though. A more structured approach, breaking them down into smaller parts that are easier to digest would have been welcome.

The evaluation is rather limited and only focuses on the number of messages sent as a function of cluster size. What would be much more interesting for me to see are actual time and bandwidth spent to complete a cluster-sending operation.

And last but not least, the quality of the writing can / should be substantially improved before publication. The paper contains many grammar mistakes and strange formulations. A non-exhaustive list of them is highlighted in the document here: https://ipfs.infura.io/ipfs/QmUBLLUDfmgkkM9Yn99Vyk5K4NfLiqj68YLvPKSPXszVya

**Expertise:**

Follow the literature closely, last published 5+ years ago

**Useful:**

yes

---

### Official Review · Reviewer_HnvD · 2022-06-17
**Review on Byzantine Cluster-sending algorithms**

**Decision:**

Weak accept: good paper with flaws that can be fixed in three months

**Review:**

The paper considers a setup with a sharded design encompassing two or more independent BFT state machine replication systems. The task is to allow the reliable transfer of quantities among two shards. The "cluster-sending-problem".

The paper introduces the problem, existing solutions and then proceeds to present and analyse their own probabilistic solution and its associated bounds in terms of number of steps. The first solution CPp selects with replacement pairs of replicas in each cluster and repeatedly tries to attain a successive transfer among them, selecting new contact replicas if failing. The second solution CSpl establishes a list of replica pairs and goes trough them (intuitively avoid repeated independent selection of faulty replicas, something possible in the prior first scheme). The paper argues how these approaches fare under synchrony, asynchrony and message faults and how it improves over other approaches in a fault free setting.

To my eyes, the paper does a fair comparison with the related work they present and shows clear improvements over it. In general presentation and readability are good.

Detailed comments:

Page 1:

"One wat" -> "One way"

When considering the sharded design to overcome limitations on consensus scalability it might be relevant to also consider that consensus might not be needed to quantity transfer tasks: cf: https://hal.archives-ouvertes.fr/hal-02861511v3

Page 6:

In the CSpl protocol, it looks that a possible optimization would order the lists of pairs, instead of fully at random, by the likelihood of them having non faulty replicas. Apparently this could favour from transfer to transfer trying first pairs that are more likely to succeed by being successful in previous transfers. Would there be any obvious drawback in this?

Page 10:

On section 6, point 2, talking about eventual occurrence of reliable communication. The concept of "Fair-lossy link" can be useful here. cf: https://dcl.epfl.ch/site/_media/education/da18-introduction.pdf

Page 11:

Figure 7 would be more clear by using in the y axis the average number of steps, instead of the total number of steps for 10000 repetitions.

Page 12:

"We assume that communication within each cluster is *reliable*. In this case, we only included our probabilistic cluster-sending protocols as PBS-CS and CHAINSPACE both assume reliable communication ...". By *reliable* did you meant to say *unreliable* here?

**Expertise:**

Follow the literature closely, last published 5+ years ago

**Useful:**

yes

---

### Meta-Review · Area_Chair_C5bD · 2022-08-10

**Recommendation:** Reject
**Confidence:** 5

**Metareview:**

Dear Authors,

We regret to inform you that your submission has not been accepted for publication at this round of JSYS. As you will see at the reviews the paper's topic is of interest in the community however there were concerns on practicality as well as core algorithmic novelty compared to two-phase commit. We hope that the reviews will prove to be helpful towards your work.

Best regards, JSYS Commitee

---

### Decision · Program_Chairs · 2022-08-10

Reject